# Global Safe Sequential Learning via Efficient Knowledge Transfer

## Abstract

Sequential learning methods such as active learning and Bayesian optimization select the most informative data to learn about a task. In many medical or engineering applications, the data selection is constrained by a priori unknown safety conditions. A promising line of safe learning methods utilize Gaussian processes (GPs) to model the safety probability and perform data selection in areas with high safety confidence. However, accurate safety modeling requires prior knowledge or consumes data. In addition, the safety confidence centers around the given observations which leads to local exploration. As transferable source knowledge is often available in safety critical experiments, we propose to consider transfer safe sequential learning to accelerate the learning of safety. We further consider a pre-computation of source components to reduce the additional computational load that is introduced by incorporating source data. In this paper, we theoretically analyze the maximum explorable safe regions of conventional safe learning methods. Furthermore, we empirically demonstrate that our approach 1) learns a task with lower data consumption, 2) globally explores multiple disjoint safe regions under guidance of the source knowledge, and 3) operates with computation comparable to conventional safe learning methods.

## 1 Introduction

Despite the great success of machine learning, accessing data is a non-trivial task. One prominent approach is to consider experimental design (Lindley, 1956; Chaloner & Verdinelli, 1995; Brochu et al., 2010). In particular, active learning (AL) (Krause et al., 2008; Kumar & Gupta, 2020) and Bayesian optimization (BO) (Brochu et al., 2010; Snoek et al., 2012) resort to a sequential data selection process. The methods initiate with a small amount of data, iteratively compute an acquisition function, query new data according to the acquisition score, receive observations from the oracle, and update the belief, until the learning goal is achieved or the acquisition budget is exhausted. These learning algorithms often utilize Gaussian processes (GPs Rasmussen & Williams (2006)) as surrogate models for the acquisition computation.

In many applications such as spinal cord stimulation (Harkema et al., 2011) and robotic learning (Berkenkamp et al., 2016; Dominik Baumann et al., 2021), the algorithms must respect some a priori unknown safety concerns. One effective approach of performing safe learning is to model the safety constraints with additional GPs (Sui et al., 2015; Schreiter et al., 2015; Zimmer et al., 2018; Yanan Sui et al., 2018; Matteo Turchetta et al., 2019; Berkenkamp et al., 2020; Sergeyev et al., 2020; Dominik Baumann et al., 2021; Li et al., 2022). The algorithms initiate with given safe observations. A safe set is then defined to restrict the exploration to regions with high safety confidence. The safe set expands as the learning proceeds, and thus the explorable area grows. Safe learning is also considered in related domains such as Markov Decision Processes (Matteo Turchetta et al., 2019) and reinforcement learning (García et al., 2015).

In this paper, we focus on GPs as they are often considered the gold-standard when it comes to calibrated uncertainties. While such safe learning methods have achieved a huge impact, few challenges remain. Firstly, GP priors need to be given prior to the exploration (Sui et al., 2015; Berkenkamp et al., 2016; 2020) or fitted with initial data (note that accessing the data is expensive) (Schreiter et al., 2015; Zimmer et al., 2018; Li et al., 2022). In addition, safe learning algorithms suffer from local exploration. GPs are typically smooth

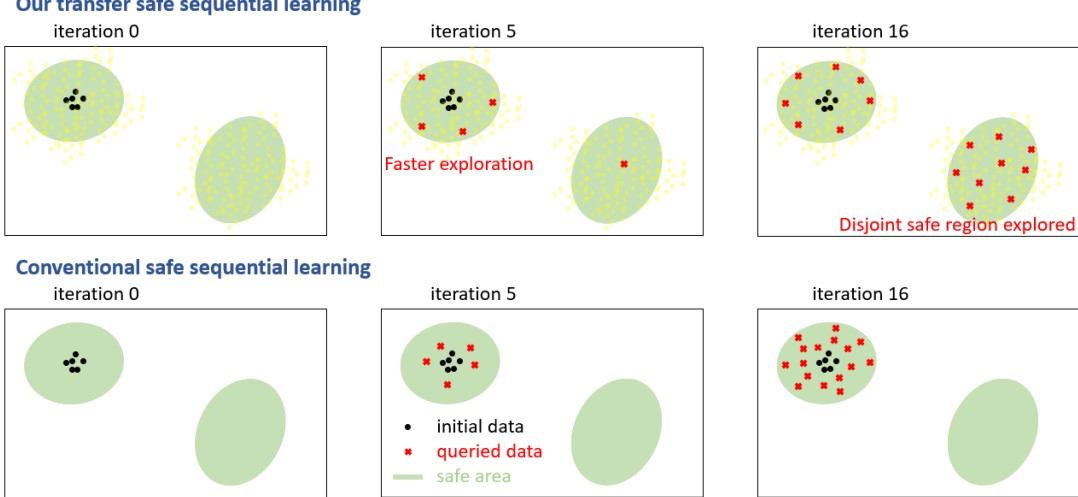

Figure 1: Illustration: safe sequential learning with transfer (top) and conventional (bottom) learning.

and the uncertainty increases beyond the reachable safe set boundary. Disconnected safe regions will be classified as unsafe and will remain unexplored. We provide a detailed analysis and illustration of explorable regions in Section 3. In reality, local exploration increases the effort of deploying safe learning algorithms because the domain experts need to provide safe data from multiple safe regions.

**Our contribution:** As safe learning (Schreiter et al., 2015; Sui et al., 2015) is always initialized with prior knowledge, we fairly assume correlated experiments have been performed and the results are available. This assumption enables transfer learning (Figure 1), where the benefit is twofold: 1) exploration as well as expansion of safe regions are significantly accelerated, and 2) the source task may provide guidance on safe regions disconnected from the initial target data and thus helps us to explore globally. Concrete applications are ubiquitous, including simulation to reality (Marco et al., 2017), serial production, and multi-fidelity modeling (Li et al., 2020).

Transfer learning can be achieved by considering the source and target tasks jointly as multi-output GPs (Journel & Huijbregts, 1976; Álvarez et al., 2012). However, GPs are notorious for the cubic time complexity due to the inversion of Gram matrices (Section 3). Large amount of source data thus introduce pronounced computational time, which is often a bottleneck in real experiments. We further modularize the multi-output GPs such that the source relevant components can be pre-computed and fixed. This alleviates the complexity of multi-output GPs while the benefit is retained.

In summary, we 1) introduce the idea of transfer safe sequential learning supported by a thorough mathematical formulation, 2) derive that conventional no-transfer approaches have an upper bound of explorable region, 3) provide a modularized approach to multi-output GPs that can alleviate the computational burden of source data, with our technique being more general than the previous method in Tighineanu et al. (2022), and 4) demonstrate the empirical efficacy.

**Related work:** Safe learning is considered in many problems such as Markov Decision Processes (Matteo Turchetta et al., 2019) and reinforcement learning (García et al., 2015). In this paper, we focus on GP learning problems. In Gelbart et al. (2014); Hernandez-Lobato et al. (2015); Hernández-Lobato et al. (2016), the authors investigated constrained learning with GPs. The authors integrated constraints directly into the acquisition function (e.g. discounting the acquisition score by the probability of constraint violation). These works do not exclude unsafe data from the search pool, and the experimenting examples are mostly not safety critical. A safe set concept was introduced for safe BO (Sui et al., 2015) and safe AL (Schreiter et al., 2015). The concept was then extended to BO with multiple safety constraints (Berkenkamp et al.,

2020), to AL for time series modeling (Zimmer et al., 2018), and to AL for multi-output problems (Li et al., 2022). For safe BO, Sui et al. Yanan Sui et al. (2018) proposed to conduct the safe set exploration and BO in two distinguished stages. All of these methods suffer from local exploration (Section 3). Sergeyev et al. (2020) considered disjoint safe regions, assuming regions separated only by a small gap where the constraint function(s), with the noise, shortly goes beneath (but still close to) the safety threshold. Dominik Baumann et al. (2021) proposed a global safe BO method on dynamical systems, assuming that unsafe areas are approached slowly enough and that there exists an intervention mechanism which stops the system quickly enough. None of these methods exploits transfer safe learning which can allow for global exploration on any systems given prior source knowledge.

Transfer learning and multi-task learning have caught increasing attention. In particular, multi-output GP methods have been developed for multi-task BO (Swersky et al., 2013; Poloczek et al., 2017), sim-to-real transfer for BO (Marco et al., 2017), and multi-task AL (Zhang et al., 2016). However, GPs have time complexity cubic to the number of observations, competed by multiple outputs. In Tighineanu et al. (2022), the authors assume a specific structure of the multi-output kernel, and factorize the computation with an ensembling technique. This eases the computational burdens for transfer sequential learning. In our paper, we propose a modularized transfer safe learning to facilitate real experiments while avoiding cubic complexity. Our modularization technique can be generalized to arbitrary multi-output kernels.

**Paper structure:** The remaining of this paper is structured as follows: we provide the goal of safe sequential learning in Section 2; in Section 3, we introduce the background and analyze the local exploration problem of safe learning; Section 4 elaborates our approach under a transfer learning scenario; Section 5 is the experimental study; finally, we conclude our paper in Section 6.

## 2 Problem statement

**Preliminary:** Throughout this paper, we inspect regression output and safety values. Each input $\boldsymbol{x} \in \mathcal{X} \subseteq \mathbb{R}^D$ has a corresponding noisy regression output $y \in \mathbb{R}$ and the corresponding noisy safety values jointly expressed as a vector $\boldsymbol{z} = (z^1, ..., z^J) \in \mathbb{R}^J$.

**Assumption 2.1.** $y = f(\boldsymbol{x}) + \epsilon_f, z^j = q_j(\boldsymbol{x}) + \epsilon_{q_j}$, where $\epsilon_f \sim \mathcal{N}\left(0, \sigma_f^2\right)$, $\epsilon_{q_j} \sim \mathcal{N}\left(0, \sigma_{q_j}^2\right)$. In addition, $y_s = f_s(\boldsymbol{x}_s) + \epsilon_{f_s}, z_s^j = q_{j,s}(\boldsymbol{x}_s) + \epsilon_{q_{j,s}}$, where $\epsilon_{f_s} \sim \mathcal{N}\left(0, \sigma_{f_s}^2\right)$, $\epsilon_{q_{j,s}} \sim \mathcal{N}\left(0, \sigma_{q_{j,s}}^2\right)$. $\{f, q_j\}$ are our target black-box function and safety functions.

The source and target tasks may have different number of safety conditions, but we can add trivial constraints (e.g. $1 \geq -\infty$) to either task in order to have the same number of constraints $J$ for both tasks. The notation is summarized in Table 1.

**Safe learning problem statement:** We are given a small number of safe observations $\mathcal{D}_{N_{init}} = \{\boldsymbol{x}_{1:N_{init}}, y_{1:N_{init}}, \boldsymbol{z}_{1:N_{init}}\}$, $\boldsymbol{x}_{1:N_{init}} = \{\boldsymbol{x}_1, ..., \boldsymbol{x}_{N_{init}}\} \subseteq \mathcal{X}$, $y_{1:N_{init}} = \{y_1, ..., y_{N_{init}}\} \subseteq \mathbb{R}$ and safety observations $\boldsymbol{z}_{1:N_{init}} := (z^1, ..., z^J)_{1:N_{init}} := (z^1_{1:N_{init}}, ..., z^J_{1:N_{init}}) = \{\boldsymbol{z}_n = (z^1_n, ..., z^J_n)\}_{n=1}^{N_{init}}$. In practice, the initial data usually meet the safety constraints, i.e. $z^j_n \geq T_j$ for all observation index $n$ and constraint index $j$. We are further given source data $\mathcal{D}_{N_{source}}^{source} = \{\boldsymbol{x}_{s,1:N_{source}}, y_{s,1:N_{source}}, \boldsymbol{z}_{s,1:N_{source}}\}$, $\boldsymbol{x}_{s,1:N_{source}} = \{\boldsymbol{x}_{s,1}, ..., \boldsymbol{x}_{s,N_{source}}\} \subseteq \mathcal{X}$, $y_{s,1:N_{source}} = \{y_{s,1}, ..., y_{s,N_{source}}\} \subseteq \mathbb{R}$ and $\boldsymbol{z}_{s,1:N_{source}} = \{\boldsymbol{z}_n = (z^1_{s,n}, ..., z^J_{s,n})|n = 1, ..., N_{source}\} \subseteq \mathbb{R}^J$. $N_{source}$ is the number of source data points. Notably, the source data do not need to be measured with the same safety constraints as the target task. In our main paper, we consider only one source task for simplicity, while Appendix E provides formulation and ablation studies on more source tasks. With one source task, we assume $N_{source}$, the number of source data, is large enough and we do not need to explore for the source task. This is often the case when there is plenty of data from previous versions of systems or prototypes.

The goal is to evaluate the function $f : \mathcal{X} \to \mathbb{R}$ where each evaluation is expensive. In each iteration, we select a point $\boldsymbol{x}_* \in \mathcal{X}_{pool} \subseteq \mathcal{X}$ to evaluate ($\mathcal{X}_{pool} \subseteq \mathcal{X}$ is the search pool which can be the entire space $\mathcal{X}$ or a predefined subspace of $\mathcal{X}$, depending on the applications). This selection should respect the a priori unknown safety constraints $\forall j = 1, ..., J, q_j(\boldsymbol{x}_*) \geq T_j$, where true $q_j$ are inaccessible. Then, a budget consuming

Table 1: Key notation

| Symbols | Meaning |
|---:|:---|
| $\mathcal{D}_N = \{\boldsymbol{x}_{1:N}, y_{1:N}, \boldsymbol{z}_{1:N}\}$ | dataset of the target task, $N = N_{init}, ..., N_{init} + num\_steps$ |
| $z_{1:N}^j = \{z_1^j, ..., z_N^J\}$ | safety observations of the $j$-th constraint (unknown function $q_j$) |
| $\boldsymbol{z}_{1:N} = (z_{1:N}^1, ..., z_{1:N}^J)$ | safety observations of all constraints jointly |
| $\mathcal{D}_{N_{source}}^{source}$ | dataset of the source task $\{\boldsymbol{x}_{s,1:N_{source}}, y_{s,1:N_{source}}, \boldsymbol{z}_{s,1:N_{source}}\}$ |
| $y = f(\boldsymbol{x}) + \epsilon_f$ | observation of unknown function $f \sim \mathcal{GP}(0, k_f), \epsilon_f \sim \mathcal{N}\left(0, \sigma_f^2\right)$ |
| $z^j = q_j(\boldsymbol{x}) + \epsilon_{q_j}$ | observation of unknown constraint $q_j \sim \mathcal{GP}(0, k_{q_j}), \epsilon_{q_j} \sim \mathcal{N}\left(0, \sigma_{q_j}^2\right)$ |
| $q_j(\boldsymbol{x}) \geq T_j$ | $j$-th safety condition |
| $y_s = f_s(\boldsymbol{x}_s) + \epsilon_{f_s}$ | source task observation prior $f_s \sim \mathcal{GP}(0, k_{f_s}), \epsilon_{f_s} \sim \mathcal{N}\left(0, \sigma_{f_s}^2\right)$ |
| $z_s^j = q_{j,s}(\boldsymbol{x}_s) + \epsilon_{q_{j,s}}$ | source task constraint prior $q_{j,s} \sim \mathcal{GP}(0, k_{q_{j,s}}), \epsilon_{q_{j,s}} \sim \mathcal{N}\left(0, \sigma_{q_{j,s}}^2\right)$ |
| $\boldsymbol{f} : \mathcal{X} \times \{\text{task indices}\} \to \mathbb{R}$ | $f_s$ and $f$ jointly as a multi-task function |
| $\boldsymbol{q}_j : \mathcal{X} \times \{\text{task indices}\} \to \mathbb{R}$ | $q_{j,s}$ and $q_j$ jointly as a multi-task function |
| $\boldsymbol{f} \sim \mathcal{GP}\left(\boldsymbol{0}, k_{\boldsymbol{f}}\right)$ | multi-task GP prior, kernel $k_{\boldsymbol{f}}$ parameterized by $\boldsymbol{\theta_f} = (\theta_{f_s}, \theta_f)$ |
| $\boldsymbol{q}_j \sim \mathcal{GP}\left(\boldsymbol{0}, k_{\boldsymbol{q}_j}\right)$ | multi-task GP prior, kernel $k_{\boldsymbol{q}_j}$ parameterized by $\boldsymbol{\theta_{q_j}} = (\theta_{q_{j,s}}, \theta_{q_j})$ |

labeling process occurs, and we obtain a noisy $y_*$ and noisy safety values $\boldsymbol{z}_*$. The labeled points are then added to $\mathcal{D}_{N_{init}}$ (observed dataset becomes $\mathcal{D}_{N_{init}+1}$), and we proceed to the next iterations (Algorithm 1). In the following, $N$ will be the size of observed dataset of the target task, and it varies from $N_{init}$ to $N_{init} + num\_steps$ (number of AL steps, i.e. AL budget). The notation is summarized in Table 1.

This problem formulation applies to both AL and BO. In this paper, we focus on AL problems. The goal is using the evaluations to make accurate predictions $f(\mathcal{X})$, and the points we select would favor general understanding over space $\mathcal{X}$, up to the safety constraints.

## 3 Background & local exploration of safe learning methods

In this section, we introduce GPs, safe learning algorithms for GPs, and then provide detailed analysis and illustration of the local exploration problem.

**Gaussian processes (GPs):** A GP is a stochastic process specified by a mean and a kernel function (Rasmussen & Williams, 2006; Kanagawa et al., 2018; Schoelkopf & Smola, 2002). Without loss of generality, we assume the GPs have zero mean. In addition, without prior knowledge to the data, it is common to assume the governing kernels are stationary.

**Assumption 3.1.** $g \in \{f, q_1, ..., q_J\}$, $g \sim \mathcal{GP}(0, k_g)$ and $k_g(\boldsymbol{x}, \boldsymbol{x}') := k_g(\boldsymbol{x} - \boldsymbol{x}') \leq 1$ are stationary.

Bounding the kernels by 1 provides advantages in theoretical analysis (Srinivas et al., 2012) and is not restrictive because the data are usually normalized to zero mean and unit variance.

The GP assumptions (Assumption 2.1 and Assumption 3.1) indicate that each of $\{f, q_1, ..., q_J\}$ has a predictive distribution given as the following. We write down the distribution for $f$ at a test point $\boldsymbol{x}_*$, while the distributions of $q_j$ can be obtained by replacing $f$ with $q_j$ and $y_{1:N}$ with $z_{1:N}^j$: $p\left(f(\boldsymbol{x}_*)|\boldsymbol{x}_{1:N}, y_{1:N}\right) = \mathcal{N}\left(\mu_{f,N}(\boldsymbol{x}_*), \sigma_{f,N}^2(\boldsymbol{x}_*)\right)$,

$$\begin{aligned}
\mu_{f,N}(\boldsymbol{x}_*) =: \mu_{f,N} &= k_f(\boldsymbol{x}_{1:N}, \boldsymbol{x}_*)^T \left(\boldsymbol{K}_f + \sigma_f^2 I\right)^{-1} y_{1:N}, \\
\sigma_{f,N}^2(\boldsymbol{x}_*) =: \sigma_{f,N}^2 &= k_f(\boldsymbol{x}_*, \boldsymbol{x}_*) - k_f(\boldsymbol{x}_{1:N}, \boldsymbol{x}_*)^T \left(\boldsymbol{K}_f + \sigma_f^2 I\right)^{-1} k_f(\boldsymbol{x}_{1:N}, \boldsymbol{x}_*),
\end{aligned} \tag{1}$$

where $k_f(\boldsymbol{x}_{1:N}, \boldsymbol{x}_*) = (k_f(\boldsymbol{x}_1, \boldsymbol{x}_*), ..., k_f(\boldsymbol{x}_N, \boldsymbol{x}_*)) \in \mathbb{R}^{N \times 1}$, and $\boldsymbol{K}_f \in \mathbb{R}^{N \times N}$ is a matrix with $[\boldsymbol{K}_f]_{ij} = k_f(\boldsymbol{x}_i, \boldsymbol{x}_j)$. Typically, $k_f$ is parameterized and can be fitted together with $\sigma_f^2$.

**Safe learning:** A core of safe learning methods (Sui et al., 2015; Yanan Sui et al., 2018; Berkenkamp et al., 2020; Dominik Baumann et al., 2021) is to compare the safety confidence bounds with the thresholds and define a safe set $\mathcal{S}_N \subseteq \mathcal{X}_{pool}$ as

$$\mathcal{S}_N = \cap_{j=1}^{J} \{\boldsymbol{x} \in \mathcal{X}_{pool} | \mu_{q_j,N}(\boldsymbol{x}) - \beta^{1/2}\sigma_{q_j,N}(\boldsymbol{x}) \geq T_j\}, \tag{2}$$

where $\beta \in \mathbb{R}^+$ is a parameter for probabilistic tolerance control (Sui et al., 2015; Berkenkamp et al., 2020). This definition is equivalent to $\forall \boldsymbol{x} \in \mathcal{S}_N, p(q_1(\boldsymbol{x}) \geq T_1, ..., q_J(\boldsymbol{x}) \geq T_J) \geq (1-\alpha)^J$ when $\alpha = 1 - \Phi(\beta^{1/2})$ (Schreiter et al., 2015; Zimmer et al., 2018; Li et al., 2022).

In each iteration, a new point is queried by mapping safe candidate inputs to acquisition scores:

$$\boldsymbol{x}_* = \operatorname{argmax}_{\boldsymbol{x} \in \mathcal{S}_N} a(\boldsymbol{x}|\mathcal{D}_N), \tag{3}$$

where $\mathcal{D}_N$ is the current observed dataset and $a$ is an acquisition function.

**Remark 3.2.** Notably, solving such a constrained optimization is challenging. In the literature (Schreiter et al., 2015; Zimmer et al., 2018; Li et al., 2022; Sui et al., 2015; Berkenkamp et al., 2020), this is solved on discrete pool with finite elements, i.e. $N_{pool} := |\mathcal{X}_{pool}| < \infty$. One would apply Equation (1) on the entire pool $\mathcal{X}_{pool}$ to determine the safe set, then optimize the acquisition scores over the safe set.

In our paper, we inherit this finite discrete pool setting. The whole learning process is summarized in Algorithm 1.

In AL problems, a prominent acquisition function is the predictive entropy: $a(\boldsymbol{x}|\mathcal{D}_N) = H_f[\boldsymbol{x}|\mathcal{D}_N] = \frac{1}{2}\log\left(2\pi e \sigma_{f,N}^2(\boldsymbol{x})\right)$ (Schreiter et al., 2015; Zimmer et al., 2018; Li et al., 2022). We use $a(\boldsymbol{x}|\mathcal{D}_N) = \sum_{g \in \{f,q_1,...,q_J\}} H_g[\boldsymbol{x}|\mathcal{D}_N]$ to accelerate the exploration of safety models. It is possible to exchange the acquisition function by SafeOpt criteria for safe BO problems (Sui et al., 2015; Berkenkamp et al., 2020; Rothfuss et al., 2022)).

---

**Algorithm 1** Sequential Learning

**Require:** $\mathcal{D}_{N_{init}}, \mathcal{X}_{pool}, \beta$ or $\alpha$
1: **for** $N = N_{init}, ..., N_{init} + num\_steps$ **do**
2:      Fit GPs $(k_f, k_{q_j}, \sigma_f^2, \sigma_{q_j}^2)$
3:      $\boldsymbol{x}_* \leftarrow \operatorname{argmax}_{\boldsymbol{x} \in \mathcal{S}_N} a(\boldsymbol{x}|\mathcal{D}_N)$
4:      Evaluate at $\boldsymbol{x}_*$ to get $y_*$ and $\boldsymbol{z}_*$
5:      $\mathcal{D}_{N+1} \leftarrow \mathcal{D}_N \cup \{\boldsymbol{x}_*, y_*, \boldsymbol{z}_*\}, \mathcal{X}_{pool} \leftarrow \mathcal{X}_{pool} \setminus \{\boldsymbol{x}_*\}$
6: **end for**

---

**Safe learning suffer from local exploration:** In this section, we analyze the upper bound of explorable safe regions. Commonly used stationary kernels (Assumption 3.1) measure the difference of a pair of points while the actual point values do not matter. These kernels have the property that closer points correlate strongly while distant points result in small kernel values. We first formulate this property as the following assumption.

**Assumption 3.3.** Given a kernel function $k : \mathcal{X} \times \mathcal{X} \to \mathbb{R}$, assume $\forall \delta > 0, \exists r > 0$ s.t. $\|\boldsymbol{x} - \boldsymbol{x}'\| \geq r \Rightarrow k(\boldsymbol{x}, \boldsymbol{x}') \leq \delta$ under $L2$ norm.

We provide expression of popular stationary kernels (RBF kernel and Matérn kernels), as well as their $r - \delta$ relations in the Appendix B.3.

In the following, we derive a theorem showing that standard kernels only allow local exploration of safety regions. The main idea is: when a point $\boldsymbol{x}_*$ is far away from the observations, we can get very small $\delta$ (i.e. small covariance measured by kernel). Thus the prediction at $\boldsymbol{x}_*$ is weakly correlated to the observations. As a result, the predictive mean is close to zero and the predictive uncertainty is large, both of which imply that the method has small safety confidence, i.e. $p\left((q_j(\boldsymbol{x}_*) \geq T_j)|\boldsymbol{x}_{1:N}, z_{1:N}^j\right)$. Here we assume that $q_j \geq T_j$ is not a trivial condition. In other words, $T_j$ is in sensitive domain of $q_j$ (i.e. $T_j$ is not far away from zero).

**Theorem 3.4** (Local exploration of single-output GPs)**.** *We are given $\forall \boldsymbol{x}_* \in \mathcal{X}$, $\boldsymbol{x}_{1:N} \subseteq \mathcal{X}$, a kernel $k_{q_j}$ satisfying Assumption 3.3 (distant points result in weak correlation) and $k_{q_j}(\cdot, \cdot) \leq 1$. Denote $k^j_{scale} := \max k_{q_j}(\cdot, \cdot)$. $q_j \sim \mathcal{GP}(0, k_{q_j})$ is a GP, $z^j_{1:N} := (z^j_1, ..., z^j_N)$ is a set of observed noisy values (Assumption 2.1) and $\|(z^j_1, ..., z^j_N)\| \leq \sqrt{N}$. Then $\forall \delta \in (0, \sqrt{k^j_{scale}} \sigma_{q_j}/\sqrt{N}), \exists r > 0$ s.t. when $\min_{\boldsymbol{x}_i \in \boldsymbol{x}_{1:N}} \|\boldsymbol{x}_* - \boldsymbol{x}_i\| \geq r$, the probability thresholded on a constant $T_j$ is bounded by $p\left((q_j(\boldsymbol{x}_*) \geq T_j)|\boldsymbol{x}_{1:N}, z^j_{1:N}\right) \leq \Phi\left(\frac{N\delta/\sigma^2_{q_j} - T_j}{\sqrt{k^j_{scale} - (\sqrt{N}\delta/\sigma_{q_j})^2}}\right)$.*

Our theorem (proof in the Appendix B.4) provides the maximum safety probability of a point as a function of its distance to the observed data in $\mathcal{X}$. Therefore, it measures an upper bound of explorable safe area. Notice that $\|z^j_{1:N}\| \leq \sqrt{N}$ is not very restrictive because an unit-variance dataset has $\|z^j_{1:N}\| = \sqrt{N}$. This theorem indicates that a standard GP with commonly used kernels explores only neighboring regions of the initial $\boldsymbol{x}_{1:N}$.

**Remark 3.5.** In Section 4, we will see that our new transfer safe sequential learning framework may explore beyond the neighborhood of target $\boldsymbol{x}_{1:N}$, taken the source input $\boldsymbol{x}_{s,1:N_{source}}$ into consideration.

In the following, we plug exact numbers into Theorem 3.4 for an illustration.

**Example 3.6.** We consider a one-dimensional toy dataset visualized in Figure 2. Assume $N = 10$, $\sigma^2_q = 0.01$ and $T = 0$. We omit $j$ because $J = 1$ here. $\sigma_q/\sqrt{N}$ is roughly 0.0316. In this example, the generated data have $\|z_{1:N}\| \leq \sqrt{10}$. We train an unit-variance ($k_{scale} = \max k_q(\cdot, \cdot) = 1$) Matérn -5/2 kernel on this example, and we obtain lengthscale $\approx 0.1256$. This kernel is strictly decreasing, so Assumption 3.3 is satisfied. In particular, $r = 4.485 * 0.1256 = 0.563316 \Rightarrow \delta \leq 0.002$, noticing that $\delta = 0.002 \Rightarrow \Phi\left(\frac{N\delta/\sigma^2_q - T}{\sqrt{1 - (\sqrt{N}\delta/\sigma_q)^2}}\right) \approx \Phi(2)$.

When the safety tolerance is set to $\beta^{1/2} = 2$, we can thus know from Theorem 3.4 that safe regions that are 0.563316 further from the observed ones are always identified as unsafe and is not explorable. In Figure 2, the two safe regions are more than 0.7 distant from each other, indicating that the right safe region is never explored by conventional safe learning methods. Please see Appendix B for numerical details and additional illustrations.

Our probability bound $\Phi\left(\frac{N\delta/\sigma^2_q - T}{\sqrt{k^j_{scale} - (\sqrt{N}\delta/\sigma_q)^2}}\right)$ is the worst case obtained with very mild assumptions. Empirically, the explorable regions found by GP models are smaller (see Figure 2 and appendix Figure 5).

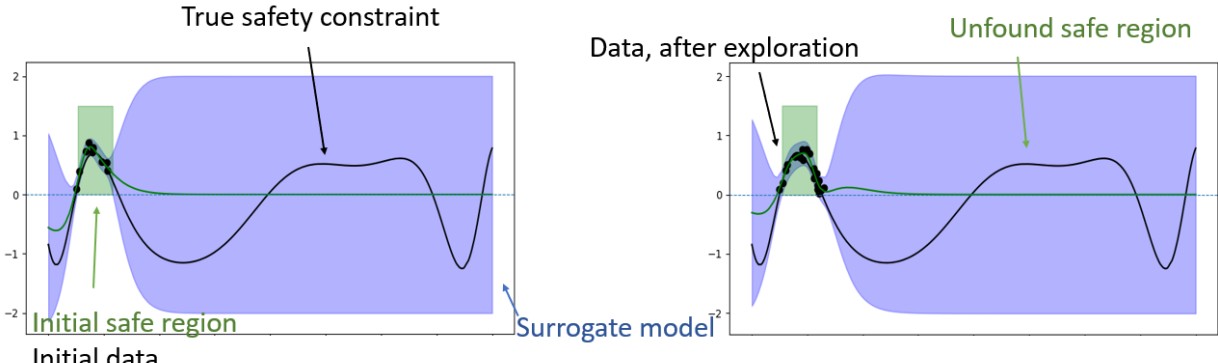

Figure 2: The safety function $q(x) = \sin\left(10x^3 - 5x - 10\right) + \frac{1}{3}x^2 - \frac{1}{2}$. The observations are with noise drawn from $\mathcal{N}(0, 0.1^2)$.

# 4 Modularized GP transfer learning

In the previous section, we introduced GP safe learning technique, and we analyzed the local exploration problem. In this section, we present our transfer learning strategy, where the aim is to facilitate safe learning and to enable global exploration if properly guided by the source data.

**Modeling the data with source knowledge:** The idea is to extend the GPs (Assumption 3.1) to multi-output models (Journel & Huijbregts, 1976; Álvarez et al., 2012; Tighineanu et al., 2022). We say $index_s$ and $index_t$ are source and target task index variables. For example, $index_t = 0$ can be target task, $index_s = 1, ...$ can be indices of multiple source tasks. In our main paper, we consider one source task, so the task indices $index_s, index_t$ are just binary. Scenarios of more source tasks are provided in Appendix E. We concatenate the source and target tasks and then define notation $\boldsymbol{f} : \mathcal{X} \times index\_space \to \mathbb{R}$ and $\boldsymbol{q}_j : \mathcal{X} \times index\_space \to \mathbb{R}$, where $\boldsymbol{f}(\cdot, index_s) = f_s(\cdot)$, $\boldsymbol{f}(\cdot, index_t) = f(\cdot)$, $\boldsymbol{q}_j(\cdot, index_s) = q_{j,s}(\cdot)$ and $\boldsymbol{q}_j(\cdot, index_t) = q_j(\cdot)$. Please also see Table 1 for the summary of our notation. The multi-task functions can then be modeled with GP as well.

**Assumption 4.1.** $\boldsymbol{f} \sim \mathcal{GP}(\boldsymbol{0}, k_{\boldsymbol{f}})$ and $\boldsymbol{q}_j \sim \mathcal{GP}(\boldsymbol{0}, k_{\boldsymbol{q}_j})$ for some stationary kernels $k_{\boldsymbol{f}}, k_{\boldsymbol{q}_j} : (\mathcal{X} \times index\_space) \times (\mathcal{X} \times index\_space) \to \mathbb{R}$.

Let $\hat{\boldsymbol{x}}_{s,1:N_{source}} := \{(\boldsymbol{x}_{s,i}, index_s) | \boldsymbol{x}_{s,i} \in \boldsymbol{x}_{s,1:N_{source}}\}$ and $\hat{\boldsymbol{x}}_{1:N} := \{(\boldsymbol{x}_i, index_t) | \boldsymbol{x}_i \in \boldsymbol{x}_{1:N}\}$ denote the input data concatenated with the task indices. Then for $\boldsymbol{g} \in \{\boldsymbol{f}, \boldsymbol{q}_j\}$, the predictive distribution given in Equation (1) becomes (similarly, we write down the distribution for $\boldsymbol{f}$, while distributions for $\boldsymbol{q}_j$ can be obtained by replacing $\boldsymbol{f}$ with $\boldsymbol{q}_j$ and $y$. with $z^j$.)

$$
\begin{aligned}
\mu_{\boldsymbol{f},N}(\boldsymbol{x}_*, index_t) &= \boldsymbol{v}_f^T \Omega_{\boldsymbol{f}}^{-1} \begin{pmatrix} y_{s,1:N_{source}} \\ y_{1:N} \end{pmatrix}, \\
\sigma_{\boldsymbol{f},N}^2(\boldsymbol{x}_*, index_t) &= k_{\boldsymbol{f}}((\boldsymbol{x}_*, index_t), (\boldsymbol{x}_*, index_t)) - \boldsymbol{v}_{\boldsymbol{f}}^T \Omega_{\boldsymbol{f}}^{-1} \boldsymbol{v}_{\boldsymbol{f}}, \\
\boldsymbol{v}_f &= k_{\boldsymbol{f}}\left(\begin{pmatrix} \hat{\boldsymbol{x}}_{s,1:N_{source}} \\ \hat{\boldsymbol{x}}_{1:N} \end{pmatrix}, (\boldsymbol{x}_*, index_t)\right) \\
\Omega_{\boldsymbol{f}} &= \begin{pmatrix} K_{f_s} + \sigma_{f_s}^2 I_{N_{source}} & K_{f_s,f} \\ K_{f_s,f}^T & K_f + \sigma_f^2 I_N \end{pmatrix}
\end{aligned}
\tag{4}
$$

where $K_{f_s} = k_{\boldsymbol{f}}(\hat{\boldsymbol{x}}_{s,1:N_{source}}, \hat{\boldsymbol{x}}_{s,1:N_{source}})$, $K_{f_s,f} = k_{\boldsymbol{f}}(\hat{\boldsymbol{x}}_{s,1:N_{source}}, \hat{\boldsymbol{x}}_{1:N})$ and $K_f = k_{\boldsymbol{f}}(\hat{\boldsymbol{x}}_{1:N}, \hat{\boldsymbol{x}}_{1:N})$. The GP model $\boldsymbol{f}$ (and $\boldsymbol{q}_j$) is governed by the multitask kernel $k_{\boldsymbol{f}}$ (and $k_{\boldsymbol{q}_j}$ for each safety function) and noise parameters $\sigma_{f_s}^2, \sigma_f^2$ (and $\sigma_{q_{j,s}}^2, \sigma_{q_j}^2$) which can be fitted with observations.

**Remark 4.2.** In our paper, we assume all safety constraints are independent. If this is not the case, one may still model different safety constraints with our notation: for example, we have 3 unknown constraints $q_1, q_2, q_3$, and the corresponding source $q_{1,s}, q_{2,s}, q_{3,s}$, then Assumption 4.1 still holds if the index space is expanded to source indices $index_s = 0, 1, 2$ specifying data of $q_{1,s}, q_{2,s}$ or $q_{3,s}$ and target indices $index_t = 3, 4, 5$ specifying $q_1, q_2$ or $q_3$, which means we model all the 6 functions jointly with one multi-output constraint function.

In this formulation, the covariance bound $\delta$ in Theorem 3.4 takes the source input $\boldsymbol{x}_{s,1:N_{source}}$ into consideration. Thus, comparing to modeling solely with target task, incorporating a source task provides the potential to significantly enlarge the area with high safety confidence (i.e. region not bounded by Theorem 3.4). We show empirically in Section 5 that global exploration is indeed easier to achieve with appropriate $\boldsymbol{x}_{s,1:N_{source}}$.

**In-experiment speed-up via source pre-computation:** Computation of $\Omega_{\boldsymbol{f}}^{-1}$ (and $\Omega_{\boldsymbol{q}_j}$) has cubic complexity $\mathcal{O}((N_{source} + N)^3)$ in time, for $N = N_{init}, ..., N_{init} + num\_steps$. This computation is also required for fitting the models: common fitting techniques include Type II ML, Type II MAP and Bayesian treatment (Snoek et al., 2012; Riis et al., 2022) over kernel and noise parameters (Rasmussen & Williams, 2006), all of which involves computing the marginal likelihoods $\mathcal{N}\left(\begin{pmatrix} y_{s,1:N_{source}} \\ y_{1:N} \end{pmatrix} | \boldsymbol{0}, \Omega_{\boldsymbol{f}}\right)$ and $\mathcal{N}\left(\begin{pmatrix} z_{s,1:N_{source}}^j \\ z_{1:N}^j \end{pmatrix} | \boldsymbol{0}, \Omega_{\boldsymbol{q}_j}\right)$. In our paper, Bayesian treatment is not considered because MC sampling is time consuming.

The goal now is to avoid calculating $\Omega_{\boldsymbol{f}}^{-1}$ and $\Omega_{\boldsymbol{q}_j}^{-1}$ repeatedly in the experiments. For brevity, we describe how we do this with $\Omega_{\boldsymbol{f}}^{-1}$, while the same principle applies to $\Omega_{\boldsymbol{q}_j}^{-1}$. For GP models, the inversion is achieved

---

**Algorithm 2** Modularized SL

---

**Require:** $\mathcal{D}_{N_{source}}^{source}, \mathcal{D}_{N_{init}}, \mathcal{X}_{pool}, \beta$ or $\alpha$
1: Fit GPs and then fix $\theta_{f_s}, \theta_{q_{j,s}}, \sigma_{f_s}, \sigma_{q_{j,s}}$
2: Compute and fix $L_{f_s}, L_{q_{j,s}}$
3: **for** $N = N_{init}, ..., N_{init} + num\_steps$ **do**
4:     Fit GPs (remaining parameters $\theta_f, \theta_{q_j}, \sigma_f, \sigma_{q_j}$)
5:     $\boldsymbol{x}_* \leftarrow \text{argmax}_{\boldsymbol{x} \in \mathcal{S}_N} a(\boldsymbol{x}|\mathcal{D}_N)$
6:     Evaluate at $\boldsymbol{x}_*$ to get $y_*$ and $\boldsymbol{z}_*$
7:     $\mathcal{D}_{N+1} \leftarrow \mathcal{D}_N \cup \{\boldsymbol{x}_*, y_*, \boldsymbol{z}_*\}, \mathcal{X}_{pool} \leftarrow \mathcal{X}_{pool} \setminus \{\boldsymbol{x}_*\}$
8: **end for**

---

by performing a Cholesky decomposition $L(\Omega_{\boldsymbol{f}})$, i.e. $\Omega_{\boldsymbol{f}} = L(\Omega_{\boldsymbol{f}})L(\Omega_{\boldsymbol{f}})^T$, where $L(\Omega_{\boldsymbol{f}})$ is a lower triangular matrix (Rasmussen & Williams, 2006), and then for any matrix $C$, $L(\Omega_{\boldsymbol{f}})^{-1}C$ is computed by solving a linear system.

We propose to perform the cholesky decomposition in two steps, as described below. The aim here is to compute part of $L(\Omega_{\boldsymbol{f}})$ beforehand. The key idea is to cluster the parameters of $k_f$ into $\boldsymbol{\theta_f} = (\theta_{f_s}, \theta_f)$, where the source $k_{\boldsymbol{f}}((\cdot, index_s), (\cdot, index_s))$ is independent of $\theta_f$. Then, as $\boldsymbol{x}_{s,1:N_{source}}$ is invariant, $K_{f_s}$ adapts only to $\theta_{f_s}$. Given that the source tasks are well explored, the source likelihood $p(y_{s,1:N_{source}}|\boldsymbol{x}_{s,1:N_{source}}) = \mathcal{N}(y_{s,1:N_{source}}|\boldsymbol{0}, K_{f_s} + \sigma_{f_s}^2 I_{N_{source}})$ can be barely increased while we explore for the target task. Thus we assume $K_{f_s}$ (i.e. $\theta_{f_s}$) and $\sigma_{f_s}^2$ remain fixed in the experiments, this allows us to isolate the source relevant computations, as the source relevant block (top left block) of $L(\Omega_{\boldsymbol{f}})$ is also fixed. We can then prepare a safe learning experiment with pre-computed $L_{f_s} = L(K_{f_s} + \sigma_{f_s}^2 I_{N_{source}})$. The same procedure applies to each $\boldsymbol{q}_j$. The learning procedure is summarized in Algorithm 2. In each iteration (line 4 of Algorithm 2), the time complexity becomes $\mathcal{O}(N_{source}^2 N) + \mathcal{O}(N_{source}N^2) + \mathcal{O}(N^3)$ instead of $\mathcal{O}\left((N_{source} + N)^3\right)$. We provide mathematical details in the Appendix C. Our technique can be applied to any multi-output kernel because the clustering $\boldsymbol{\theta_f} = (\theta_{f_s}, \theta_f)$ does not require independence of $k_{\boldsymbol{f}}((\cdot, index_s), (\cdot, index_t))$ and $k_{\boldsymbol{f}}((\cdot, index_t), (\cdot, index_t))$ from $\theta_{f_s}$. The same principle applies to $\boldsymbol{q}_j$.

**Kernel selection:** In the following, we briefly review existing multi-output GP models and motivate selection of the model we use later in our experiments. Here, each $\boldsymbol{g} \in \{\boldsymbol{f}, \boldsymbol{q}_1, ..., \boldsymbol{q}_J\}$ is a multi-output GP correlating source and target tasks. The task indices are binary: $index_s = 0$ is source and $index_t = 1$ is target. A widely investigated multi-output framework is the linear model of corregionalization (LMC): $k_{\boldsymbol{g}} = \sum_l \begin{pmatrix} W_{l,s}^2 + \kappa_s & W_{l,s}W_{l,t} \\ W_{l,s}W_{l,t} & W_{l,t}^2 + \kappa \end{pmatrix} \otimes k_l(\cdot, \cdot)$, i.e. a $2 \times 2$ matrix specified by task indices, where $k_l(\cdot, \cdot)$ is a standard kernel as in Assumption 3.1, and $(W_l W_l^T + diag(\kappa_s, \kappa))$ learns the task correlation induced by the $l$-th latent function (Álvarez et al., 2012). Here, each $\boldsymbol{g}$ has its own kernel, but we omit $\boldsymbol{g}$ in the parameter subscripts for brevity. When pairing this kernel with our Algorithm 2, we observe that the training can become unstable due to multiple local optima in the first phase (line 1 of Algorithm 2). This may be because LMC learns joint patterns from all present tasks.

In Poloczek et al. (2017); Marco et al. (2017); Tighineanu et al. (2022), the authors consider a hierarchical GP (HGP): $k_{\boldsymbol{g}} = \begin{pmatrix} k_s(\cdot, \cdot) & k_s(\cdot, \cdot) \\ k_s(\cdot, \cdot) & k_s(\cdot, \cdot) + k_t(\cdot, \cdot) \end{pmatrix}$. Similarly, each $\boldsymbol{g}$ has its own kernel, but we omit $\boldsymbol{g}$ in the parameter subscripts for brevity. HGP is a variant of LMC, where the target task is treated as a sum of the source (modeled by $k_s$) and the target-source residual (modeled by $k_t$). This formulation has the benefit that the fitting of source ($k_s$) and residual ($k_t$) are separated and thus makes HGP a good model to run Algorithm 2 (set $\theta_{g_s}$ the parameters of $k_s$ and $\theta_{g_s}$ the parameters of $k_t$).

In Tighineanu et al. (2022), the authors derived an ensembling technique allowing also for a source pre-computation. Their technique is equivalent to our method when we use HGP, but our approach can be generalized to any multi-output kernels (with implicit restriction that a source fitting of the chosen model needs to be accurate) while the ensembling technique is limited to HGP.

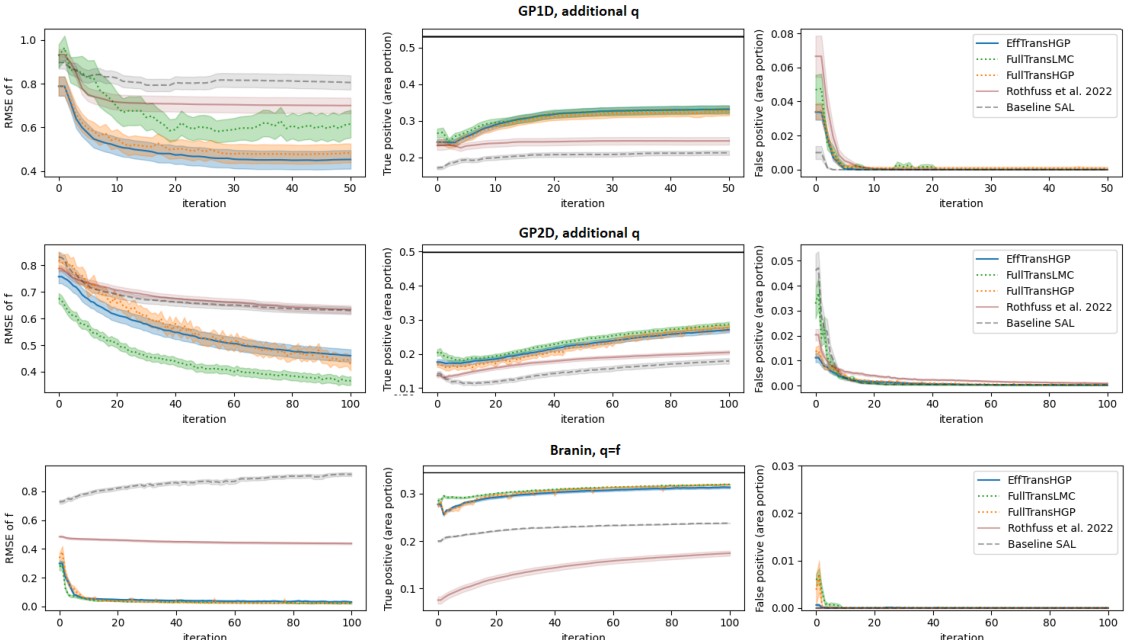

Figure 3: Safe AL experiments on three benchmark datasets. GP data: $f$ and safety function $q \geq 0$ over $\mathcal{X} = [-2, 2]^D$, $D = 1$ ($N_{source} = 50$, $N_{init} = 10$, 50 data points are queried) or $D = 2$ ($N_{source} = 250$, $N_{init} = 20$, 100 data points are queried). Branin data: constraint $q = f \geq 0$ (Section 5.1), $N_{source} = 100$, $N_{init} = 20$, 100 data points are queried. The results are mean and one standard error of 100 (GP data) or 25 (Branin data) experiments. The test points for RMSEs are sampled from all of the true safe area, including the regions individual methods (e.g. SAL) may fail to explore. Note that FullTransLMC has more than ten model parameters, while in GP1D dataset we start with $N = 10$. The TP/FP safe areas are portion of the input space area. Ground true safe area portion of each dataset is marked black in the second column. Please also see appendix Figure 10 for fitting time and region cluster of each query.

In our experiments, we perform Algorithm 2 with HGP as our main pipeline, and Algorithm 1 with LMC (more flexible in learning yet slow) and with HGP as full transfer scenarios. The base kernels $k_s, k_t, k_l$ are all Matérn-5/2 kernel with $D$ lengthscale parameters ($\mathcal{X} \subseteq \mathbb{R}^D$). The scaling variance of $k_l$ is fixed to 1 because it can be absorbed into the output-covariance terms (see above). One can of course change the base kernel as long as it is suitable for the application. Although we did not pair Algorithm 2 with LMC as discussed above, note that our modularized computation scheme can still benefit the general LMC in closely related settings, e.g. (i) datasets in which more than one source task is available or (ii) sequential learning schemes that only refit the GPs after receiving a batch of query points.

## 5 Experiments

In this section, we perform safe AL experiments to answer the following questions: **1)** do multi-output GPs facilitate learning of disconnected safe regions, **2)** is it more data efficient to learn with transfer safe learning than applying a conventional method, and **3)** how is the runtime of our modularized approach compared with the baseline?

We compare five experimental setups: **1)** EffTransHGP: Algorithm 2 with multi-output HGP, **2)** FullTransHGP: Algorithm 1 with multi-output HGP, **3)** FullTransLMC: Algorithm 1 with multi-output LMC, **4)** Rothfuss et al. 2022: GP model meta learned with the source data by applying Rothfuss et al. (2022), and **5)** SAL: the conventional Algorithm 1 with single-output GPs and Matérn-5/2 kernel.

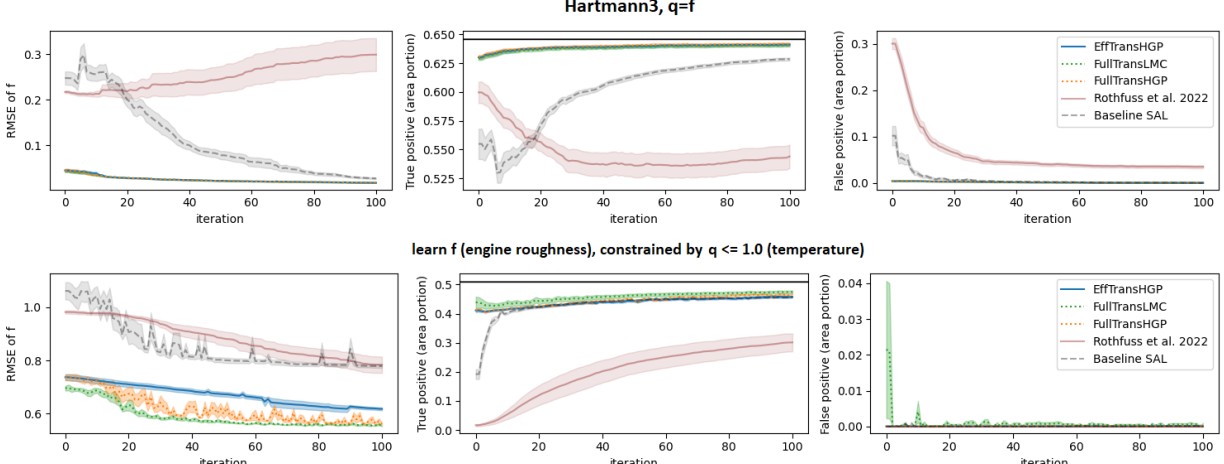

Figure 4: Safe AL experiments on Hartmann3 and engine modeling problems. Hartmann3: $N_{source} = 100$, $N$ is from 20 to 120, results are mean and one standard error of 25 experiments. Engine: $N_{source} = 500$, $N$ is from 20 to 120, results are mean and one standard error of 5 repetitions.

Table 2: Number of discovered regions

| methods | GP1D+z | GP2D+z | Branin |
|---:|:---:|:---:|:---:|
| *num_steps* | 50 | 100 | 100 |
| EffTransHGP | $1.79 \pm 0.07$ | $2.77 \pm 0.13$ | $2 \pm 0$ |
| FullTransHGP | $1.78 \pm 0.07$ | $3 \pm 0.14213$ | $2 \pm 0$ |
| FullTransLMC | $1.78 \pm 0.08$ | $2.68 \pm 0.14$ | $2 \pm 0$ |
| Rothfuss2022 | $1.22 \pm 0.05$ | $1.07 \pm 0.03$ | $1 \pm 0$ |
| SAL | $1 \pm 0$ | $1.29 \pm 0.09$ | $1 \pm 0$ |

Transfer learning discovers multiple disjoint safe regions while baselines stick to neighborhood of the initial region. In appendix Figure 10, we track the number of explored regions per iteration.

For the safety tolerance, we always fix $\beta = 4$, i.e. $\alpha = 1 - \Phi(\beta^{1/2}) = 0.02275$ (Equation (2)), implying that each fitted GP safety model allows 2.275% unsafe tolerance when inferring safe set Equation (2). Notice that with Rothfuss et al. (2022), the GP model parameters are trained up-front and remain fixed during the experiments. Rothfuss et al. 2022 considered safe BO problems. We change the acquisition function to entropy so it becomes a safe AL method. Our code will be published on GitHub.

**Test problems with tractable safe regions:** We start from simple simulated problems with input dimension $D = 1$ or $D = 2$ (GP1D, GP2D, Branin problems). In such cases, it is analytically and computationally possible to cluster the disconnected safe regions via connected component labeling (CCL) algorithms (He et al., 2017). This means, in each iteration of the experiments, we track to which safe region each observation belongs (Table 2). In these initial experiments, we generate one source dataset and one target dataset such that the target task has at least two disjoint safe regions, each of which has a portion also safe in the source problem. The design is due to the selection of our kernels. Our base kernel, the Matérn-5/2 kernel, correlates closeness of data points, and LMC and HGP rescale the Matérn-5/2 kernel measures for different tasks, which means patterns of the same area in the space are transferred. Modeling more complicated transferring pattern, e.g. correlation on an abstract feature space, may require a careful selection of appropriate base kernel (see e.g. Bitzer et al. (2022)). In the main experiments, $N_{source}$ (the

Table 3: Training time

| methods $(N_{source}, N)$ | GP1D+z $(100, 10 + 50)$ | GP2D+z $(250, 20 + 100)$ | Branin $(100, 20 + 100)$ | Hartmann3 $(100, 20 + 100)$ | Engine $(500, 20 + 100)$ |
|---|---|---|---|---|---|
| EffTransHGP | $8.947 \pm 0.198$ | $10.73 \pm 0.190$ | $3.754 \pm 0.121$ | $3.662 \pm 0.089$ | $9.596 \pm 0.418$ |
| FullTransHGP | $9.171 \pm 0.133$ | $39.31 \pm 0.639$ | $8.129 \pm 0.267$ | $9.092 \pm 0.467$ | $124.99 \pm 5.608$ |
| FullTransLMC | $26.56 \pm 0.628$ | $202.8 \pm 12.43$ | $21.16 \pm 1.207$ | $34.43 \pm 1.664$ | $615.7 \pm 27.99$ |
| Rothfuss2022 | $0.0 \pm 0.0$ | $0.0 \pm 0.0$ | $0.0 \pm 0.0$ | $0.0 \pm 0.0$ | $0.0 \pm 0.0$ |
| SAL | $6.881 \pm 0.083$ | $8.044 \pm 0.142$ | $4.691 \pm 0.078$ | $4.073 \pm 0.083$ | $4.686 \pm 0.243$ |

The training time (s) of $\boldsymbol{f}$ and $\boldsymbol{q}$ (if $\boldsymbol{q}$ is not $\boldsymbol{f}$) at the last iteration.

number of source data points) is fixed for each problem. In our Appendix E, we provide ablation studies on the Branin dataset, where we vary the number of source data points and number of source tasks.

**General test problem and real-world problem:** We first consider a problem with higher dimension: Hartmann3 ($D = 3$). Here, it becomes computationally not tractable to cluster the safe regions. Thus (i) the source task, the source data and the initial target data are all sampled randomly (in contrast to GP1D, GP2D and Branin where we focus on problems with disjoint safe regions), and (ii) the CCL algorithm, i.e. the safe region clustering, is not performed during the AL experiments. Secondly, an engine modeling problem is considered, where we transfer from one engine dataset to another. This problem (i) has noisy grid values interpolated from raw measurements which makes the CCL algorithm inaccurate, and (ii) the safe set of this target task is not clearly separated into multiple disjoint regions. Therefore, the CCL algorithm is not performed during the experiments.

**Metrics:** The learning result of $\boldsymbol{f}$ is shown as RMSEs between the GP mean prediction and test $y$ sampled from true safe regions. To measure the performance of $\boldsymbol{q}$, we use the area of $\mathcal{S}_N$ (Equation (2)), as this indicates the explorable coverage of the space. In particular, we look at the area of $\mathcal{S}_N \cap \mathcal{S}_{true}$ (true positive or TP area, the larger the better) and $\mathcal{S}_N \cap (\mathcal{X} \setminus \mathcal{S}_{true})$ (false positive or FP area, the smaller the better). Here, $\mathcal{S}_{true} \subseteq \mathcal{X}_{pool}$ is the set of true safe candidate inputs, and this is available since our datasets in the experiments are prepared as executed queries. With GP1D, GP2D and Branin data, CCL (He et al., 2017) is performed to cluster which safe region each query belongs to (Table 2).

## 5.1 AL on problems with tractable safe regions

**Datasets:** We adapt algorithm 1 of Kanagawa et al. (2018) to generate multi-output GP samples. The first output is treated as our source task and the second output as the target task. We have one main function $\boldsymbol{f}$ and an additional safety function $\boldsymbol{q}$. Numerical details and example datasets are plotted in Appendix D. We generate 10 datasets and repeat the AL experiments five times for each dataset. For Branin data, we take the numerical setting from Rothfuss et al. (2022); Tighineanu et al. (2022) to generate five different datasets. With each dataset, we repeat the experiments for five times.

**Result:** In Figure 3, we show the results of GP1D, GP2D and of Branin data. We see that EffTransHGP, FullTransHGP and FullTransLMC experiments achieve accurate and much larger safe set coverage (larger TP area and small FP area). In addition, the learning of $\boldsymbol{f}$ is more efficient with EffTransHGP, FullTransHGP and FullTransLMC as the RMSE drops faster compared to the baseline methods. Note that the test points are sampled from all of the true safe area, including the part baseline SAL fails to explore. It is thus not guaranteed that RMSE of SAL monotonically decreases (Branin). We observe from the experiments that the meta learning approach, Rothfuss et al. 2022, fails to generalize to larger area, which might be due to a lack of data in target task representativeness (one source, very few for meta learning) or/and in quantity.

In Table 2, we count the number of safe regions explored by the queries. This confirms the ability to explore disjoint safe regions. One remark is that Branin function is smooth and has two clear safe regions; while huge stochasticity exists in GP data and we may have various number of small or large safe regions scattered

in the space. Table 3 shows the model fitting time, confirming that EffTransHGP has comparable time complexity as baseline SAL, as opposed to FullTransHGP and FullTransLMC. We provide additional ratios of safe queries in appendix Table 4, which is a sanity check that the methods are indeed safe.

Please note the learning flexibility is FullTransLMC > FullTransHGP > EffTransHGP, and our experimental results are consistent to this intuition (RMSE of FullTransLMC in 1D data is worse because we starts with 10 data points which is less than the number of LMC parameters, Figure 3).

### 5.2 AL on general test problem and real-world problem

**Hartmann problem:** We take the numerical setting from Rothfuss et al. (2022); Tighineanu et al. (2022) to generate five different Hartmann3 datasets. With each dataset, we repeat the experiments for five times. Please see Appendix D.2 for details. In this experiment, EffTransHGP, FullTransLMC and FullTransHGP provide much smaller RMSEs and larger safe area (Figure 4).

**Engine datasets:** We have two datasets, measured from the same prototype of engine under different conditions. Both datasets measure the temperature, roughness, emission HC, and emission NOx. The raw data were measured by operating an engine and the measurement equipments. We perform independent AL experiments to learn about roughness (Figure 4) and temperature (put in appendix Figure 11), both constrained by the normalized temperature values $q \leq 1.0$. The safe set is around 0.5293 of the entire space. The datasets have two free variables and two contextual inputs which are supposed to be fixed. The contextual inputs are recorded with noise, so we interpolate the values with a multi-output GP simulator, trained on the full datasets. Thus this experiment is performed on a semi-simulated condition. Details are given in Appendix D.2.

The safe set of this target task is not clearly separated into multiple disjoint regions. Thus the conventional method can eventually identify most part of the safe area. Nevertheless, we still see a much better RMSEs and much less data consumption for large safe set coverage (Figure 4). We also observe that Rothfuss et al. 2022 failed to generalize the meta-learned source knowledge to the entire target space exploration.

## 6 Conclusion

We propose a transfer safe sequential learning to facilitate real experiments. We demonstrate its pronounced acceleration of learning which can be seen by a faster drop of RMSE and a larger safe set coverage. At the same time, our modularized multi-output modeling 1) retains the potential of performing global GP safe learning and 2) alleviates the cubic complexity in the experiments, leading to a considerable reduce of time complexity.

**Limitations:** Our modularized method is in theory compatible with any multi-output kernel, in contrast to the ensemble technique in Tighineanu et al. (2022) which is only valid for a specific kernel. However, one limitation of source precomputation is that it requires to fix correct source relevant hyperparameters solely with source data (e.g. HGP is a good candidate due to its separable source-target structure while LMC, which learns joint patterns of tasks, will not be fixed correctly with only source data). Another limitation is that the benefit of transfer learning relies on multi-task correlation. This means transfer learning will not be helpful when the correlation is absent, or when the source data are not present in our target safe area. Modeling with more complicated base kernel (we use Matérn-5/2 kernel) may enable more sophisticated multi-task correlations, but this is beyond the scope of this paper (see e.g. Bitzer et al. (2022) for kernel selections).

## Acknowledgements

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

# A  Appendix Overview

Appendix B provides detailed analysis and illustrations of our main theorem. In Appendix C, we demonstrate the math of our source pre-computation technique. Appendix D contains the experiment details and Appendix E the ablation studies, additional plots and tables.

# B  GPs with classical stationary kernels cannot jump through an unsafe valley

## B.1  Bound of explorable region of safe learning methods

In our main script, we provide a bound of the safety probability. The theorem is restated here.

**Theorem 3.3.** We are given $\forall \boldsymbol{x}_* \in \mathcal{X}$, $\boldsymbol{x}_{1:N} \subseteq \mathcal{X}$, a kernel $k_{q_j}$ satisfying Assumption 3.3 and $k_{q_j}(\cdot, \cdot) \leq 1$. Denote $k_{scale}^j := max\ k_{q_j}(\cdot, \cdot)$. $q_j \sim \mathcal{GP}(0, k_{q_j})$ is a GP, $z_{1:N}^j := (z_1^j, ..., z_N^j)$ is a set of observed noisy values (Assumption 2.1) and $\|(z_1^j, ..., z_N^j)\| \leq \sqrt{N}$. Then $\forall \delta \in (0, \sqrt{k_{scale}^j} \sigma_{q_j} / \sqrt{N})$, $\exists r > 0$ s.t. when $\min_{\boldsymbol{x}_i \in \boldsymbol{x}_{1:N}} \|\boldsymbol{x}_* - \boldsymbol{x}_i\| \geq r$, the probability thresholded on a constant $T_j$ is bounded by $p\left((q_j(\boldsymbol{x}_*) \geq T_j)|\boldsymbol{x}_{1:N}, z_{1:N}^j\right) \leq \Phi\left(\frac{N\delta/\sigma_{q_j}^2 - T_j}{\sqrt{k_{scale}^j - (\sqrt{N}\delta/\sigma_{q_j})^2}}\right)$.

In this section, we illustrate a concrete example of our theorem, where conventional methods cannot explore the entire safe set in the space. Then we provide the proof of this theorem.

## B.2  Single-output GP does not reach disconnected safe region

We plug some exact numbers into the probability bound. Consider an one dimensional situation as Figure 2 and Figure 5. We omit $j$ because $J = 1$ here. Assume

1. $N = 10$,

2. $\sigma_q^2 = 0.01$,

3. $T = 0$ (notice $z_{1:N}^j$ is normalized to 0-mean and unit-variance).

In this example, the generated data have $\|z_{1:N}\| \leq \sqrt{N}$ (see Figure 2 for the rough functional values). Noticed also that $\sigma_q / \sqrt{N}$ is around 0.0316. We fix $k_{scale} := max\ k_q(\cdot, \cdot) = 1$ (the surrogate model in Figure 2). Then our theoretical bound of the safety probability is $\Phi\left(\frac{N\delta/\sigma^2 - T}{\sqrt{1 - (\sqrt{N}\delta/\sigma)^2}}\right) = \Phi\left(\frac{1000\delta}{\sqrt{1 - 1000\delta^2}}\right)$.

In our main script, $\boldsymbol{x}_*$ is unsafe if $p\left((q_j(\boldsymbol{x}_*) \geq T_j)|\boldsymbol{x}_{1:N}, z_{1:N}^j\right) < 1 - \Phi(-\beta^{1/2}) = \Phi(\beta^{1/2})$. We set the safety tolerance to $\beta^{1/2} = 2$. The decision boundary of our theorem $\frac{1000\delta}{\sqrt{1 - 1000\delta^2}} = 2$ means $\delta \approx 0.002$.

From Appendix B.3 we see that $\|\boldsymbol{x} - \boldsymbol{x}'\| \geq 4.485 \Rightarrow \delta \leq 0.002$ for unit lengthscale Matérn-5/2 kernel. With a lengthscale parameter $l$, this becomes $\frac{\|\boldsymbol{x} - \boldsymbol{x}'\|}{l} \geq 4.485 \Leftrightarrow \|\boldsymbol{x} - \boldsymbol{x}'\| \geq 4.485 * l$. Therefore $\delta \leq 0.002$ if $\|\boldsymbol{x} - \boldsymbol{x}'\| \geq 4.485 * l$.

The GP model trained on this example has lengthscale $\approx 0.1256$ (the surrogate model in Figure 2 and in left of Figure 5), so points that are at least $4.485 * 0.1256 = 0.563316$ away from the observations are always identified unsafe. Thus the safe region on the right is never inferred as safe and is not explored with conventional single-output GP model ( Figure 5, left), because the distance between the two disjoint safe regions is around 0.7. We also show empirically that a multi-output GP model transfer safety confidence from a source task and identify safe region $\mathcal{S}_{sub2}$( Figure 5, right).

## B.3  $r$-$\delta$ relation for commonly used kernels

Our main theorem consider kernels satisfying Assumption 3.2 which is restated here:

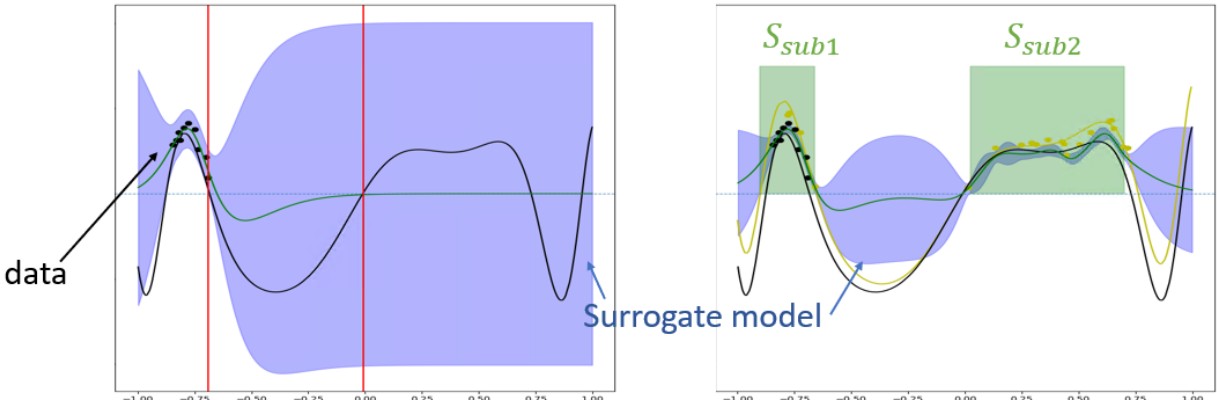

Figure 5: The safety function $q(x) = \sin\left(10x^3 - 5x - 10\right) + \frac{1}{3}x^2 - \frac{1}{2}$. Safety threshold is set to $T = 0$. The observations are with noise drawn from $\mathcal{N}(0, 0.01)$. Left: a GP with Matérn-5/2 kernel (lengthscale $\approx 0.1256$) is shown. The red lines indicate the largest observed $\boldsymbol{x}$ and the closest safe point of another region. The gap between the red lines is close to 0.7, which is beyond explorable region of conventional safe learning methods. Right: the multi-output model uses an LMC kernel with 2 latent Matérn-5/2 kernels (Álvarez et al., 2012). Additional noisy data from function $q_s(x) = \sin\left(10x^3 - 5x - 10\right) + \sin(x^2) - \frac{1}{2}$ are provided (yellow). $\mathcal{S}_{sub1}$ and $\mathcal{S}_{sub2}$ are the safe set inferred by the LMC.

**Assumption 3.2.** Given a kernel function $k : \mathcal{X} \times \mathcal{X} \to \mathbb{R}$, assume $\forall \delta > 0$, $\exists r > 0$ s.t. $\|\boldsymbol{x} - \boldsymbol{x}'\| \geq r \Rightarrow k(\boldsymbol{x}, \boldsymbol{x}') \leq \delta$ under $L2$ norm.

Notice that this assumption is weaker than $k$ being strictly decreasing (see e.g. Lederer et al. (2019)), and it does not explicitly force stationarity.

Here we want to find the exact $r$ for commonly used kernels, given a $\delta$. The following kernels (denoted by $k(\cdot, \cdot)$) are described in their standard forms. In the experiments, we often add a lengthscale $l$ and variance $k_{scale}$, i.e. $k_{parameterized}(\boldsymbol{x}, \boldsymbol{x}') = k_{scale}k(\boldsymbol{x}/l, \boldsymbol{x}'/l)$ where $k_{scale}$ and $l$ are trainable parameters. The lengthscale $l$ can also be a vector, where each component is a scaling factor of the corresponding dimension of the data.

**RBF kernel**
$k(\boldsymbol{x}, \boldsymbol{x}') = \exp\left(-\|\boldsymbol{x} - \boldsymbol{x}'\|^2/2\right)$:
$k(\boldsymbol{x}, \boldsymbol{x}') \leq \delta \Leftrightarrow \|\boldsymbol{x} - \boldsymbol{x}'\| \geq \sqrt{\log \frac{1}{\delta^2}}$.

E.g. $\delta \leq 0.3 \Leftarrow \|\boldsymbol{x} - \boldsymbol{x}'\| \geq 1.552$
$\delta \leq 0.1 \Leftarrow \|\boldsymbol{x} - \boldsymbol{x}'\| \geq 2.146$
$\delta \leq 0.002 \Leftarrow \|\boldsymbol{x} - \boldsymbol{x}'\| \geq 3.526$

**Matérn-1/2 kernel**
$k(\boldsymbol{x}, \boldsymbol{x}') = \exp\left(-\|\boldsymbol{x} - \boldsymbol{x}'\|\right)$: $k(\boldsymbol{x}, \boldsymbol{x}') \leq \delta \Leftrightarrow \|\boldsymbol{x} - \boldsymbol{x}'\| \geq \log \frac{1}{\delta}$.

E.g. $\delta \leq 0.3 \Leftarrow \|\boldsymbol{x} - \boldsymbol{x}'\| \geq 1.204$
$\delta \leq 0.1 \Leftarrow \|\boldsymbol{x} - \boldsymbol{x}'\| \geq 2.303$
$\delta \leq 0.002 \Leftarrow \|\boldsymbol{x} - \boldsymbol{x}'\| \geq 6.217$

**Matérn-3/2 kernel**
$k(\boldsymbol{x}, \boldsymbol{x}') = \left(1 + \sqrt{3}\|\boldsymbol{x} - \boldsymbol{x}'\|\right) \exp\left(-\sqrt{3}\|\boldsymbol{x} - \boldsymbol{x}'\|\right)$:

E.g. $\delta \le 0.3 \Leftarrow \|\boldsymbol{x} - \boldsymbol{x}'\| \ge 1.409$
$\delta \le 0.1 \Leftarrow \|\boldsymbol{x} - \boldsymbol{x}'\| \ge 2.246$
$\delta \le 0.002 \Leftarrow \|\boldsymbol{x} - \boldsymbol{x}'\| \ge 4.886$

**Matérn-5/2 kernel**
$k(\boldsymbol{x}, \boldsymbol{x}') = \left(1 + \sqrt{5}\|\boldsymbol{x} - \boldsymbol{x}'\| + \frac{5}{3}\|\boldsymbol{x} - \boldsymbol{x}'\|^2\right) \exp\left(-\sqrt{5}\|\boldsymbol{x} - \boldsymbol{x}'\|\right)$:

E.g. $\delta \le 0.3 \Leftarrow \|\boldsymbol{x} - \boldsymbol{x}'\| \ge 1.457$
$\delta \le 0.1 \Leftarrow \|\boldsymbol{x} - \boldsymbol{x}'\| \ge 2.214$
$\delta \le 0.002 \Leftarrow \|\boldsymbol{x} - \boldsymbol{x}'\| \ge 4.485$

### B.4   Proof of our main theorem

We first introduce some necessary theoretical properties in Appendix B.4.1, and then use the properties to prove Theorem 3.3 in Appendix B.4.2.

### B.4.1   Additional lemmas

**Definition B.1.** Let $k : \mathcal{X} \times \mathcal{X} \to \mathbb{R}$ be a kernel, $\boldsymbol{A} \subseteq \mathcal{X}$ be any dataset of finite number of elements, and let $\sigma$ be any positive real number, denote $\Omega_{k,\boldsymbol{A},\sigma^2} := k(\boldsymbol{A}, \boldsymbol{A}) + \sigma^2 I$.

**Definition B.2.** Given a kernel $k : \mathcal{X} \times \mathcal{X} \to \mathbb{R}$, dataset $\boldsymbol{A} \subseteq \mathcal{X}$, and some positive real number $\sigma$, then for $\boldsymbol{x} \in \mathcal{X}$, the $k$-, $\boldsymbol{A}$-, and $\sigma^2$-dependent function $\boldsymbol{h}(\boldsymbol{x}) = k(\boldsymbol{A}, \boldsymbol{x})^T \Omega_{k,\boldsymbol{A},\sigma^2}^{-1}$ is called a weight function (Silverman, 1984).

**Proposition B.3.** $C \in \mathbb{R}^{M \times M}$ is a positive definite matrix and $\boldsymbol{b} \in \mathbb{R}^M$ is a vector. $\lambda_{max}$ is the maximum eigenvalue of $C$. We have $\|C\boldsymbol{b}\|_2 \le \lambda_{max}\|\boldsymbol{b}\|_2$.

*Proof of Proposition B.3.*
Because $C$ is positive definite (symmetric), we can find orthonormal eigenvectors $\{\boldsymbol{e}_1, ..., \boldsymbol{e}_M\}$ of $C$ that form a basis of $\mathbb{R}^M$. Let $\lambda_i$ be the eigenvalue corresponding to $\boldsymbol{e}_i$, we have $\lambda_i > 0$.

As $\{\boldsymbol{e}_1, ..., \boldsymbol{e}_M\}$ is a basis, there exist $b_1, ..., b_M \in \mathbb{R}$ s.t. $\boldsymbol{b} = \sum_{i=1}^M b_i \boldsymbol{e}_i$. Since $\{\boldsymbol{e}_i\}$ is orthonormal, $\|\boldsymbol{b}\|_2^2 = \sum_i b_i^2$. Then

$$\|C\boldsymbol{b}\|_2 = \|\sum_{i=1}^M b_i \lambda_i \boldsymbol{e}_i\|_2 = \sqrt{\sum_{i=1}^M b_i^2 \lambda_i^2}$$
$$\le \sqrt{\sum_{i=1}^M b_i^2 \lambda_{max}^2} = \lambda_{max} \sqrt{\sum_{i=1}^M b_i^2} = \lambda_{max}\|\boldsymbol{b}\|_2$$

. 

$\square$

**Proposition B.4.** $\forall \boldsymbol{A} \subseteq \mathcal{X}$, any kernel $k$, and any positive real number $\sigma$, an eigenvalue $\lambda$ of $\Omega_{k,\boldsymbol{A},\sigma^2}$ (Definition B.1) must satisfy $\lambda \ge \sigma^2$.

*Proof of Proposition B.4.*
Let $\boldsymbol{K} := k(\boldsymbol{A}, \boldsymbol{A})$. We know that

1. $\boldsymbol{K}$ is positive semidefinite, so it has only non-negative eigenvalues, denote the minimal one by $\lambda_K$, and

2. $\sigma^2$ is the only eigenvalue of $\sigma^2 I$.

Then Weyl's inequality immediately gives us the result: $\lambda \geq \lambda_K + \sigma^2 \geq \sigma^2$. $\qquad\square$

**Corollary B.5.** *We are given* $\forall \boldsymbol{x}_* \in \mathcal{X}$, $\boldsymbol{A} \subseteq \mathcal{X}$, *any kernel $k$ satisfying Assumption 3.2 and any positive real number $\sigma$. Let $M \coloneqq$ number of elements of $\boldsymbol{A}$, and let $\boldsymbol{B} \in \mathbb{R}^M$ be a vector. Then $\forall \delta > 0, \exists r > 0$ s.t. when $\min_{\boldsymbol{x}' \in \boldsymbol{A}} \|\boldsymbol{x}_* - \boldsymbol{x}'\| \geq r$, we have*

1. $|\boldsymbol{h}(\boldsymbol{x}_*)\boldsymbol{B}| \leq \sqrt{M}\delta\|\boldsymbol{B}\|/\sigma^2$ *(see also Definition B.2),*

2. $k(\boldsymbol{x}_*, \boldsymbol{x}_*) - k(\boldsymbol{A}, \boldsymbol{x}_*)^T \Omega_{k,\boldsymbol{A},\sigma^2}^{-1} k(\boldsymbol{A}, \boldsymbol{x}_*) \geq k(\boldsymbol{x}_*, \boldsymbol{x}_*) - M\delta^2/\sigma^2$ *(see also Definition B.1).*

*Proof of Corollary B.5.*
Let $\boldsymbol{K} \coloneqq k(\boldsymbol{A}, \boldsymbol{A})$.

Proposition B.4 implies that the eigenvalues of $\left(\boldsymbol{K} + \sigma^2 I\right)^{-1}$ are bounded by $\frac{1}{\sigma^2}$.

In addition, $\min_{\boldsymbol{x}' \in \boldsymbol{A}} \|\boldsymbol{x}_* - \boldsymbol{x}'\| \geq r \Rightarrow$ all components of row vector $k(\boldsymbol{x}_*, \boldsymbol{A})$ are in region $[0, \delta]$.

1. Apply Cauchy-Schwarz inequality (line 1) and Proposition B.3 (line 2), we obtain

$$\left| k(\boldsymbol{A}, \boldsymbol{x}_*)^T \left(k(\boldsymbol{A}, \boldsymbol{A}) + \sigma^2 I\right)^{-1} \boldsymbol{B} \right| \leq \|k(\boldsymbol{A}, \boldsymbol{x}_*)^T\| \| \left(\boldsymbol{K} + \sigma^2 I\right)^{-1} \boldsymbol{B}\|$$
$$\leq \|k(\boldsymbol{A}, \boldsymbol{x}_*)\| \frac{1}{\sigma^2} \|\boldsymbol{B}\|$$
$$\leq \|(\delta, ..., \delta)\| \frac{1}{\sigma^2} \|\boldsymbol{B}\|$$
$$\leq \frac{\sqrt{M}\delta\|\boldsymbol{B}\|}{\sigma^2}.$$

2. $\left(\boldsymbol{K} + \sigma^2 I\right)^{-1}$ is positive definite Hermititian matrix, so

$$k(\boldsymbol{A}, \boldsymbol{x}_*)^T \left(\boldsymbol{K} + \sigma^2 I\right)^{-1} k(\boldsymbol{A}, \boldsymbol{x}_*) \leq \frac{1}{\sigma^2} \|k(\boldsymbol{A}, \boldsymbol{x}_*)\|^2$$
$$\leq \frac{1}{\sigma^2} M\delta^2.$$

Then, we immediately see that

$$k(\boldsymbol{x}_*, \boldsymbol{x}_*) - k(\boldsymbol{A}, \boldsymbol{x}_*)^T \left(\boldsymbol{K} + \sigma^2 I\right)^{-1} k(\boldsymbol{A}, \boldsymbol{x}_*) \geq k(\boldsymbol{x}_*, \boldsymbol{x}_*) - \frac{1}{\sigma^2} \|k(\boldsymbol{A}, \boldsymbol{x}_*)\|^2$$
$$\geq k(\boldsymbol{x}_*, \boldsymbol{x}_*) - \frac{1}{\sigma^2} M\delta^2.$$

$\qquad\square$

**Remark B.6.** A CDF of a standard Gaussian distribution is often denoted by $p(x \leq T) = \Phi(T), x \sim \mathcal{N}(0, 1)$. Notice that $p(x \leq -T) = \Phi(-T) = 1 - \Phi(T) = p(x \geq T)$.

### B.4.2 Main proof

**Theorem 3.3.** *We are given* $\forall \boldsymbol{x}_* \in \mathcal{X}$, $\boldsymbol{x}_{1:N} \subseteq \mathcal{X}$, *a kernel $k_{q_j}$ satisfying Assumption 3.3 and $k_{q_j}(\cdot, \cdot) \leq 1$. Denote $k_{scale}^j \coloneqq max\ k_{q_j}(\cdot, \cdot)$. $q_j \sim \mathcal{GP}(0, k_{q_j})$ is a GP, $z_{1:N}^j \coloneqq (z_1^j, ..., z_N^j)^T$ is a set of observed noisy values (Assumption 2.1) and $\|(z_1^j, ..., z_N^j)\| \leq \sqrt{N}$. Then $\forall \delta \in (0, \sqrt{k_{scale}^j}\sigma_{q_j}/\sqrt{N}), \exists r > 0$ s.t. when $\min_{\boldsymbol{x}_i \in \boldsymbol{x}_{1:N}} \|\boldsymbol{x}_* - \boldsymbol{x}_i\| \geq r$, the probability thresholded on a constant $T_j$ is bounded by*
$$p\left((q_j(\boldsymbol{x}_*) \geq T_j)|\boldsymbol{x}_{1:N}, z_{1:N}^j\right) \leq \Phi\left(\frac{N\delta/\sigma_{q_j}^2 - T_j}{\sqrt{k_{scale}^j - (\sqrt{N}\delta/\sigma_{q_j})^2}}\right).$$

*Proof.*
From Equation (1) in the main script, we know that

$$p\left(q_j(\boldsymbol{x}_*)|\boldsymbol{x}_{1:N}, z_{1:N}^j\right) = \mathcal{N}\left(\boldsymbol{x}_*|\mu_{q_j,N}(\boldsymbol{x}_*), \sigma_{q_j,N}^2(\boldsymbol{x}_*)\right)$$

$$\mu_{q_j,N}(\boldsymbol{x}_*) = k_{q_j}(\boldsymbol{x}_{1:N}, \boldsymbol{x}_*)^T \left(k_{q_j}(\boldsymbol{x}_{1:N}, \boldsymbol{x}_{1:N}) + \sigma_{q_j}^2 I_N\right)^{-1} z_{1:N}^j$$

$$\sigma_{q_j,N}^2(\boldsymbol{x}_*) = k_{q_j}(\boldsymbol{x}_*, \boldsymbol{x}_*) - k_{q_j}(\boldsymbol{x}_{1:N}, \boldsymbol{x}_*)^T \left(k_{q_j}(\boldsymbol{x}_{1:N}, \boldsymbol{x}_{1:N}) + \sigma_{q_j}^2 I_N\right)^{-1} k_{q_j}(\boldsymbol{x}_{1:N}, \boldsymbol{x}_*).$$

We also know that (Remark B.6)

$$p\left((q_j(\boldsymbol{x}_*) \geq T_j)|\boldsymbol{x}_{1:N}, z_{1:N}^j\right) = 1 - \Phi\left(\frac{T_j - \mu_{q_j,N}(\boldsymbol{x}_*)}{\sigma_{q_j,N}(\boldsymbol{x}_*)}\right)$$

$$= \Phi\left(\frac{\mu_{q_j,N}(\boldsymbol{x}_*) - T_j}{\sigma_{q_j,N}(\boldsymbol{x}_*)}\right).$$

From Corollary B.5, we get $\frac{\mu_{q_j,N}(\boldsymbol{x}_*) - T_j}{\sigma_{q_j,N}(\boldsymbol{x}_*)} \leq \frac{\sqrt{N}\delta\|z_{1:N}^j\|/\sigma_{q_j}^2 - T_j}{\sqrt{k_{q_j}(\boldsymbol{x}_*, \boldsymbol{x}_*) - N\delta^2/\sigma_{q_j}^2}}$. This is valid because we assume $\delta <$
$\sqrt{k_{scale}^j}\sigma_{q_j}/\sqrt{N}$. Then with $\|z_{1:N}^j\| \leq \sqrt{N}$ and the fact that $\Phi$ is an increasing function, we immediately see the result

$$p\left((q_j(\boldsymbol{x}_*) \geq T_j)|\boldsymbol{x}_{1:N}, z_{1:N}^j\right) \leq \Phi\left(\frac{N\delta/\sigma_{q_j}^2 - T_j}{\sqrt{k_{scale}^j - (\sqrt{N}\delta/\sigma_{q_j})^2}}\right).$$

$\square$

## C  Multi-output GPs with source pre-computation

Given a multi-output GP $\boldsymbol{g} \sim \mathcal{GP}(0, k_{\boldsymbol{g}})$, $\boldsymbol{g} \in \{\boldsymbol{f}, \boldsymbol{q}_1, ..., \boldsymbol{q}_J\}$, where $k_{\boldsymbol{g}}$ is an arbitrary kernel, the main computational challenge is to compute the inverse or Cholesky decomposition of

$$\Omega_{\boldsymbol{g}} = \begin{pmatrix} K_{g_s} + \sigma_{g_s}^2 I_{N_{source}} & K_{g_s,g} \\ K_{g_s,g}^T & K_g + \sigma_g^2 I_N \end{pmatrix}.$$

Such computation has time complexity $\mathcal{O}\left((N_{source} + N)^3\right)$. We wish to avoid this computation repeatedly. As in our main script, $k_{\boldsymbol{g}}$ is parameterized and we write the parameters as $\boldsymbol{\theta}_{\boldsymbol{g}} = (\theta_{g_s}, \theta_g)$, where $k_{\boldsymbol{g}}\left((\cdot, index_s), (\cdot, index_s)\right)$ is independent of $\theta_g$. $k_{\boldsymbol{g}}\left((\cdot, index_s), (\cdot, index_t)\right)$ and $k_{\boldsymbol{g}}\left((\cdot, index_t), (\cdot, index_t)\right)$ does not need to be independent of $\theta_{g_s}$

Here we propose to fix $K_{g_s}$ (i.e. $\theta_{g_s}$) and $\sigma_{g_s}^2$ and precompute the Cholesky decomposition of the source components, $L_{g_s} = L(K_{g_s} + \sigma_{g_s}^2 I_{N_{source}})$, then

$$L(\Omega_{\boldsymbol{g}}) = \begin{pmatrix} L_{g_s} & \boldsymbol{0} \\ \left(L_{g_s}^{-1} K_{g_s,g}\right)^T & L\left(\hat{K}_t\right) \end{pmatrix},$$

$$\hat{K}_t = K_g + \sigma_g^2 I_N - \left(L_{g_s}^{-1} K_{g_s,g}\right)^T L_{g_s}^{-1} K_{g_s,g}. \tag{5}$$

This is obtained from the definition of Cholesky decomposition, i.e. $\Omega_{\boldsymbol{g}} = L(\Omega_{\boldsymbol{g}}) L(\Omega_{\boldsymbol{g}})^T$, and from the fact that a Cholesky decomposition exists and is unique for any positive definite matrix.

The complexity of computing $L(\Omega_{\boldsymbol{g}})$ thus becomes $\mathcal{O}(N_{source}^2 N) + \mathcal{O}(N_{source} N^2) + \mathcal{O}(N^3)$ instead of $\mathcal{O}\left((N_{source} + N)^3\right)$. In particular, computing $L_{g_s}^{-1} K_{\boldsymbol{g},st}$ is $\mathcal{O}(N_{source}^2 N)$, acquiring matrix product $\hat{K}_t$ is $\mathcal{O}(N_{source} N^2)$ and Cholesky decomposition $L(\hat{K}_t)$ is $\mathcal{O}(N^3)$.

The learning procedure is summarized in Algorithm 2 in the main script. We prepare a safe learning experiment with $\mathcal{D}_{N_{source}}^{source}$ and initial $\mathcal{D}_N$; we fix $\theta_{f_s}, \theta_{q_{j,s}}, \sigma_{f_s}, \sigma_{q_{j,s}}$ to appropriate values, and we precompute $L_{f_s}, L_{q_{j,s}}$. During the experiment, the fitting and inference of GPs (for data acquisition) are achieved by incorporating Equation (5) in Equation (4) of the main script (Section 4).

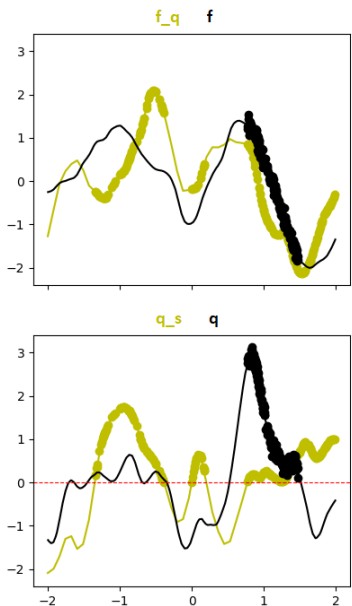

Figure 6: Example simulated GP data of $D = 1$, $\boldsymbol{f}$ is the function we want to learn (top), under an additional safety constraint $\boldsymbol{q} \geq 0$ (bottom). The curves are true source (yellow) and target (black) functions. The dots are safe source data and a pool of initial target ticket (this pool of target data are more than those actually used in the experiments).

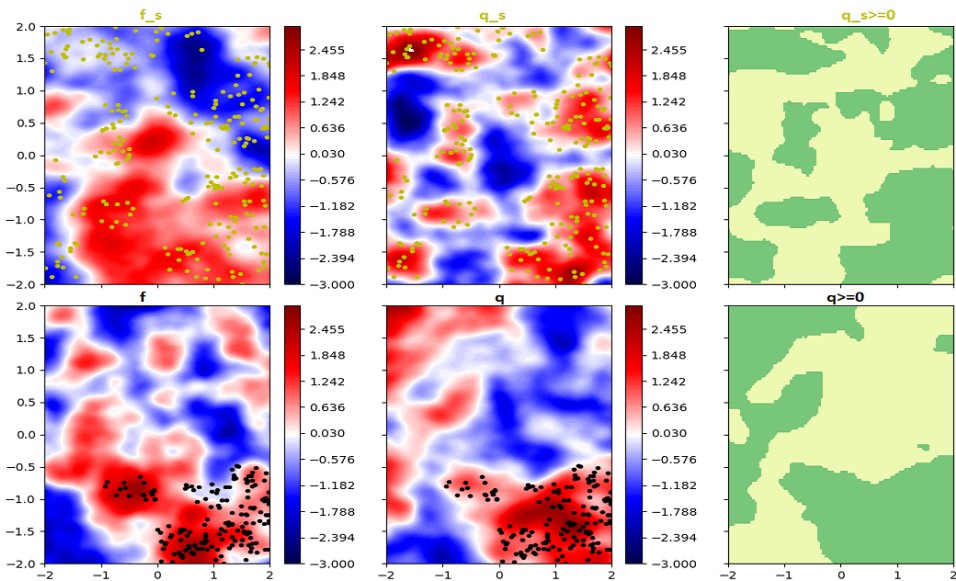

Figure 7: Example simulated GP data of $D = 2$, $\boldsymbol{f}$ is the function we want to learn (left), with an additional safety function $\boldsymbol{q}$ (middle), and the green is true safe regions $\boldsymbol{q} \geq 0$ (right). The top is source task and the bottom is target task. The dots are safe source data and a pool of initial target ticket (this pool of target data are more than those actually used in the experiments).

# D  Experiment details

## D.1  Labeling safe regions

The goal is to label disjoint safe regions, so that we may track the exploration of each land. In our experiments, the test safety values are always available because we are dealing with executed pool of data. It is thus possible to access safety conditions of each test point as a binary label. We perform connected component labeling (CCL, see He et al. (2017)) to the safety classes over grids (grids are available, see the following sections). When $D = 1$, this labeling is trivial. When $D = 2$, we consider 4-neighbors of each pixel (He et al., 2017). With simulated datasets, the ground truth is available, and thus CCL is deterministic. The CCL can be computationally intractable on high dimension (number of grids grows exponentially), and this method can be inaccurate over real data where observations are noisy and grid values need interpolation from the measurements.

After clustering the safe regions over grids, we identify which safe region each test point $\boldsymbol{x}_*$ belongs to by searching the grid nearest to $\boldsymbol{x}_*$. See main Table 2 and the queried regions count of Figure 10 for the results.

## D.2  Numerical details

When we run algorithm 1 and 2 (in the main paper), we set $N_{init}$ (number of initial observed target data), $N_{source}$ (number of observed source data) and $N_{pool}$ (size of discretized input space $\mathcal{X}_{pool}$) as follows:

1. GP1D: $N_{source} = 100$, $N_{init} = 10$, run Algorithm 1 or Algorithm 2 for 50 iterations, and $N_{pool} = 5000$;

2. GP2D: $N_{source} = 250$, $N_{init} = 20$, run Algorithm 1 or Algorithm 2 for 100 iterations, and $N_{pool} = 5000$;

3. Branin & Hartmann3: $N_{source} = 100$, $N_{init} = 20$, run Algorithm 1 or Algorithm 2 for 100 iterations, and $N_{pool} = 5000$;

4. Engine: $N_{source} = 500$, $N_{init} = 20$, run Algorithm 1 or Algorithm 2 for 100 iterations, and $N_{pool} = 3000$.

In the following, we describe in details how to prepare each dataset.

We first sample source and target test functions and then sample initial observations from the functions. With GP1D, GP2D and Branin problems (Section 5.1), we reject the sampled functions unless all of the following conditions are satisfied: (i) the target task has at least two disjoint safe regions, (ii) each of these regions has a common safe area shared with the source, and (iii) for at least two disjoint target safe regions, each aforementioned shared area is larger than 5% of the overall space (in total, at least 10% of the space is safe for both the source and the target tasks).

In our general test problems, i.e. Hartmann3 (Section 5.2), we generate functions as they are. In other words, we do not restrict the datasets to any safe region characteristics.

**GP data:**  We generate datasets of two outputs. The first output is treated as our source task and the second output as the target task.

To generate the multi-output GP datasets, we use GPs with zero mean prior and multi-output kernel $\sum_{l=1}^{2} W_l W_l^T \otimes k_l(\cdot, \cdot)$, where $\otimes$ is the Kronecker product, each $W_l$ is a 2 by 2 matrix and $k_l$ is a unit variance Matérn-5/2 kernel (Álvarez et al., 2012). All components of $W_l$ are generated in the following way: we randomly sample from a uniform distribution over interval $[-1, 1)$, and then the matrix is normalized such that each row of $W_l$ has norm 1. Each $k_l$ has an unit variance and a vector of lengthscale parameters, consisting of $D$ components. For GP1D and GP2D problems, each component of the lengthscale is sampled from a uniform distribution over interval $[0.1, 1)$. We adapt algorithm 1 of Kanagawa et al. (2018) for GP sampling, detailed as follows:

1. sample input dataset $\boldsymbol{X} \in \mathbb{R}^{n \times D}$ within interval $[-2, 2]$, and $n = 100^D$.

2. for $l = 1, 2$, compute Gram matrix $K_l = k_l(\boldsymbol{X}, \boldsymbol{X})$.

3. compute Cholesky decomposition $L_l = L(W_l W_l^T \otimes K_l) = L(W_l W_l^T) \otimes L(K_l)$ (i.e. $W_l W_l^T \otimes K_l = L_l L_l^T$, $L_l \in \mathbb{R}^{2*n \times 2*n}$).

4. for $l = 1, 2$, draw $u_l \sim \mathcal{N}(\mathbf{0}, I_{2*n})$ ($u_l \in \mathbb{R}^{(2*n) \times 1}$).

5. obtain noise-free output dataset $\boldsymbol{F} = \sum_{l=1}^{2} L_l u_l$

6. reshape $\boldsymbol{F} = \begin{pmatrix} \boldsymbol{f}(\boldsymbol{X}, s) \\ \boldsymbol{f}(\boldsymbol{X}, t) \end{pmatrix} \in \mathbb{R}^{2*n \times 1}$ into $\boldsymbol{F} = \begin{pmatrix} \boldsymbol{f}(\boldsymbol{X}, s) & \boldsymbol{f}(\boldsymbol{X}, t) \end{pmatrix} \in \mathbb{R}^{n \times 2}$.

7. normalize $\boldsymbol{F}$ again s.t. each column has mean 0 and unit variance.

8. generate initial observations (more than needed in the experiments, always sampled from the largest safe region shared between the source and the target).

During the AL experiments, the generated data $\boldsymbol{X}$ and $\boldsymbol{F}$ are treated as grids. We construct an oracle on continuous space $[-2, 2]^D$ by interpolation. During the experiments, the training data and test data are blurred with a Gaussian noise of standard deviation 0.01.

Once we sample the GP hyperparameters, we sample one main function $\boldsymbol{f}$ and an additional safety function from the GP. During the experiments, the constraint is set to $\boldsymbol{q} \geq 0$. For each dimension, we generate 10 datasets and repeat the AL experiments 5 times for each dataset. We illustrate examples of $\boldsymbol{X}$ and $\boldsymbol{F}$ in Figure 6 and Figure 7.

**Branin data:** The Branin function is a function defined over $(x_1, x_2) \in \mathcal{X} = [-5, 10] \times [0, 15]$ as

$$f_{a,b,c,r,s,t}((x_1, x_2)) = a(x_2 - bx_1^2 + cx_1 - r) + s(1 - t)cos(x_1) + s,$$

where $a, b, c, r, s, t$ are constants. It is common to set $(a, b, c, r, s, t) = (1, \frac{5.1}{4\pi^2}, \frac{5}{\pi}, 6, 10, \frac{1}{8\pi})$, which is our setting for target task.

We take the numerical setting of Tighineanu et al. (2022); Rothfuss et al. (2022) to generate five different source datasets (and later repeat 5 experiments for each dataset):

$$\begin{aligned} a &\sim Uniform(0.5, 1.5), \\ b &\sim Uniform(0.1, 0.15), \\ c &\sim Uniform(1.0, 2.0), \\ r &\sim Uniform(5.0, 7.0), \\ s &\sim Uniform(8.0, 12.0), \\ t &\sim Uniform(0.03, 0.05). \end{aligned}$$

After obtaining the constants for our experiments, we sample noise free data points and use the samples to normalize our output

$$f_{a,b,c,r,s,t}((x_1, x_2))_{normalize} = \frac{f_{a,b,c,r,s,t}((x_1, x_2)) - mean(f_{a,b,c,r,s,t})}{std(f_{a,b,c,r,s,t})}.$$

Then we set safety constraint $f \geq 0$ and sample initial safe data. The sampling noise is Gaussian during the experiments.

**Hartmann3 data:** The Hartmann3 function is a function defined over $\boldsymbol{x} \in \mathcal{X} = [0,1]^3$ as

$$f_{a_1,a_2,a_3,a_4}((x_1,x_2,x_3)) = -\sum_i^4 a_i exp\left(-\sum_{j=1}^3 A_{i,j}(x_j - P_{i,j})^2\right),$$

$$\boldsymbol{A} = \begin{pmatrix} 3 & 10 & 30 \\ 0.1 & 10 & 35 \\ 3 & 10 & 30 \\ 0.1 & 10 & 35 \end{pmatrix},$$

$$\boldsymbol{P} = 10^{-4} \begin{pmatrix} 3689 & 1170 & 2673 \\ 4699 & 4387 & 7470 \\ 1091 & 8732 & 5547 \\ 381 & 5743 & 8828 \end{pmatrix},$$

where $a_1, a_2, a_3, a_4$ are constants. It is common to set $(a_1, a_2, a_3, a_4) = (1, 1.2, 3, 3.2)$, which is our setting for target task.

We take the numerical setting of Tighineanu et al. (2022) to generate five different source datasets (and later repeat 5 experiments for each dataset):

$$a_1 \sim Uniform(1.0, 1.02),$$
$$a_2 \sim Uniform(1.18, 1.2),$$
$$a_3 \sim Uniform(2.8, 3.0),$$
$$a_4 \sim Uniform(3.2, 3.4).$$

After obtaining the constants for our experiments, we sample noise free data points and use the samples to normalize our output

$$f_{a_1,a_2,a_3,a_4}((x_1,x_2,x_3))_{normalize} = \frac{f_{a_1,a_2,a_3,a_4}((x_1,x_2,x_3)) - mean(f_{a_1,a_2,a_3,a_4})}{std(f_{a_1,a_2,a_3,a_4})}.$$

Then we set safety constraint $f \geq 0$ and sample initial safe data. The sampling noise is Gaussian during the experiments.

**Engine data** We have 2 datasets, measured from the same prototype of engine under different conditions. Both datasets measure the temperature, roughness, emission HC, and emission NOx. The inputs are engine speed, relative cylinder air charge, position of camshaft phaser and air-fuel-ratio. The contextual input variables "position of camshaft phaser" and "air-fuel-ratio" are desired to be fixed. These two contextual inputs are recorded with noise, so we interpolate the values with a multi-output GP simulator. We construct a LMC trained with the 2 datasets, each task as one output. During the training, we split each of the datasets (both safe and unsafe) into 60% training data and 40% test data. After the model parameters are selected, the trained models along with full dataset are utilized as our GP simulators (one simulator for each output channel, e.g. temperature simulator, roughness simulator, etc). The first output of each GP simulator is the source task and the second output the target task. The simulators provide GP predictive mean as the observations. During the AL experiments, the input space is a rectangle spanned from the datasets, and $\mathcal{X}_{pool}$ is a discretization of this space from the simulators with $N_{pool} = 3000$. We set $N_{source} = 500$, $N = 20$ (initially) and we query for 100 iterations ($N = 20 + 100$). When we fit the models for simulators, the test RMSEs (60% training and 40% test data) of roughness is around 0.45 and of temperature around 0.25.

In a sequential learning experiment, the surrogate models are trainable GP models. These surrogate models interact with the simulators, i.e. take $\mathcal{X}_{pool}$ from the simulators, infer the safety and query from $\mathcal{X}_{pool}$, and then obtain observations from the simulators. In our main Algorithms 1 to 2, the surrogate models are the GP models while the GP simulators are systems that respond to queries $\boldsymbol{x}_*$.

# E    Ablation Studies and Further Experiments

In this section, we provide ablation studies on the size of source dataset.

**One source task, varied** $N_{source}$**:**   We perform experiments on the Branin function. The results are presented in Figure 8. The first conclusion is that all of the multioutput methods outperform baseline safe AL (safe AL result shown in Figure 3). Note again that the RMSEs are evaluated on the entire space while the baseline safe AL explore only one safe region. In addition, we observe that more source data result in better performances, i.e. lower RMSE and larger safe set coverage (TF area), while there exist a saturation level at around $N_{source} = 100$.

**Multiple source tasks:**   Next, we wish to manipulate the number of source tasks. Before presenting the results, we first introduce the model on multiple source tasks. In this paragraph, we say $D_{source}$ is the number of source tasks. As described in Section 4, each $\boldsymbol{g} \in \{\boldsymbol{f}, \boldsymbol{q}_1, ..., \boldsymbol{q}_J\}$ is a multi-output GP correlating source and target tasks. The LMC, linear model of corregionalization, can be taken without any change: $k_{\boldsymbol{g}} = \sum_l \left(\boldsymbol{W}_l \boldsymbol{W}_l^T + diag\boldsymbol{\kappa}\right) \otimes k_l(\cdot, \cdot)$, where $k_l(\cdot, \cdot)$ is a standard single task kernel as in Assumption 3.1, and $\boldsymbol{W}_l$ and $\boldsymbol{\kappa}$ are vector of $(D_{source} + 1)$ elements (Álvarez et al., 2012). The HGP can be extended in two ways, models in Poloczek et al. (2017) or in Tighineanu et al. (2022). Here we take the model from Tighineanu et al. (2022): $k_{\boldsymbol{g}} = \sum_{i=0}^{D_{source}} Mask_i \otimes k_i(\cdot, \cdot)$, $Mask_i \in \mathbb{R}^{D_{source}+1 \times D_{source}+1}$ is a matrix where the first $i$ rows and columns are zero and the other entries are all one (all elements of $Mask_0$ are ones). One can see that if $D_{source} = 1$, then we get the HGP described in Section 4 by reindexing $k_0$ and $k_1$ here.

In this study, we generate source data with constraints, but disjoint safe regions requirement when we sample the source tasks and data (in Section 5.1, the data are generated s.t. source and target task has large enough shared safe area). We consider 1, 3 or 4 source tasks, and we generate 20 or 30 data points per task. In general, we see that 3 source tasks significantly outperform 1 source task while the performance saturates as adding 10 more points per source task seems to benefit more than adding one more source task. Note here that all source data are generated independently, i.e. the observations of each task are not restricted to the same input locations.

**Further plots and experiments:**   In Section 5.1, we track the safe region of each query in AL experiments. We measure the model fitting time per iteration as well. The main Table 2 and Table 3 present only the summary results. In Figure 10, we additionally provide the region clustering and fitting time w.r.t. AL iterations. Furthermore, Table 4 counts among the AL selected queries which, after a safety measurements are accessed, satisfy the safety constraints. This table is a sanity check that the methods are selecting points safely.

With the engine datasets, we perform additional experiments of learning $\boldsymbol{f} = \boldsymbol{q}$ =temperature, and the results are shown in Figure 11.

Table 4: Ratio of safe queries

| methods | GP1D + z | GP2D + z | Branin | Hartmann3 |
| --- | --- | --- | --- | --- |
| *num_steps* | 50 | 100 | 100 | 100 |
| EffTransHGP | $0.986 \pm 0.001$ | $0.974 \pm 0.002$ | $0.999 \pm 0.0006$ | $0.972 \pm 0.003$ |
| FullTransHGP | $0.979 \pm 0.004$ | $0.952 \pm 0.005$ | $0.9996 \pm 0.0004$ | $0.972 \pm 0.003$ |
| FullTransLMC | $0.984 \pm 0.002$ | $0.969 \pm 0.002$ | $0.993 \pm 0.0009$ | $0.968 \pm 0.003$ |
| Rothfuss2022 | $0.975 \pm 0.003$ | $0.905 \pm 0.006$ | $1.0 \pm 0.0$ | $0.84 \pm 0.011$ |
| SAL | $0.995 \pm 0.001$ | $0.958 \pm 0.005$ | $1.0 \pm 0.0$ | $0.966 \pm 0.002$ |

Ratio of all queries selected by the methods which are safe in the ground truth (initial data not included, see Section 5 for the experiments). This is a sanity check in additional to FP safe set area, demonstrates that all the methods are safe during the experiments (our datasets have 0 mean, the constraint $q \geq 0$ indicates that around half of the space is unsafe). Note: $\beta = 4$ (equivalently $\alpha = 1 - \Phi(\beta^{1/2}) = 0.002275$) implies 2.275 % unsafe tolerance is allowed by each fitted GP safety model. Engine results are not shown because the queries are all safe (the modeling FP safe set area is almost zero in this problem, see Figure 4 and Figure 11).

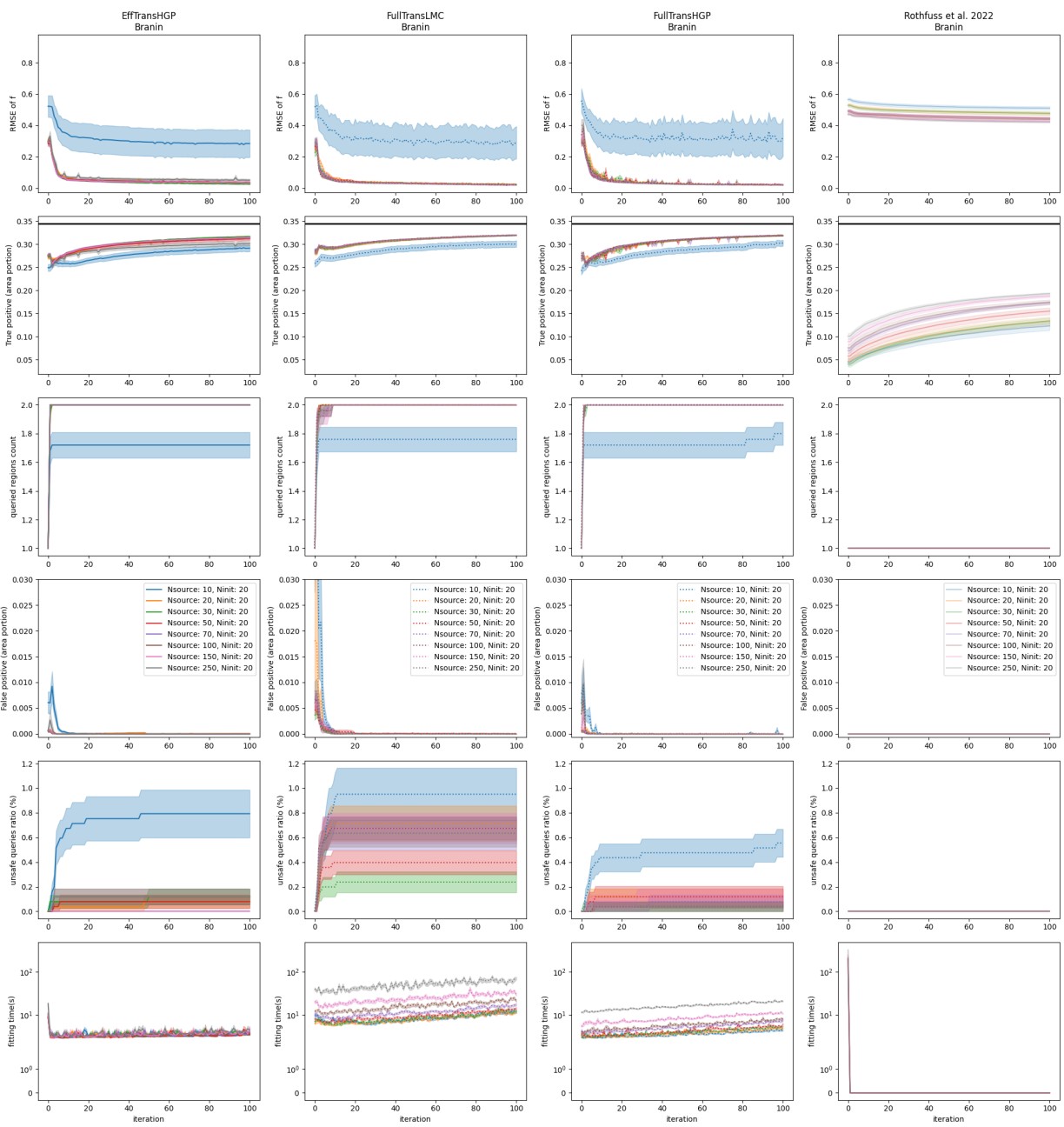

Figure 8: Safe AL experiments: Branin data with different number of source data. Each multi-task method is plotted in one column. The results are mean and one standard error of 25 experiments per setting. $\mathcal{X}_{pool}$ is discretized from $\mathcal{X}$ with $N_{pool} = 5000$. The TP/FP areas are computed as number of TP/FP points divided by $N_{pool}$ (i.e. TP/FP as portion of $\mathcal{X}_{pool}$). The third row shows the number of disjoint safe regions explored by the queries. The fifth row, the unsafe queries ratio, are presented as percentage of number of iterations (e.g. at the 2nd-iteration out of a total of 100 iterations, one of the two queries is unsafe, then the ratio is 1 divided by 100). The last row demonstrates the model fitting time. At the first iteration (iter 0-th), this includes the time for fitting both the source components and the target components (EffTransHGP). With Rothfuss et al. 2022, source fitting is the meta learning phase.

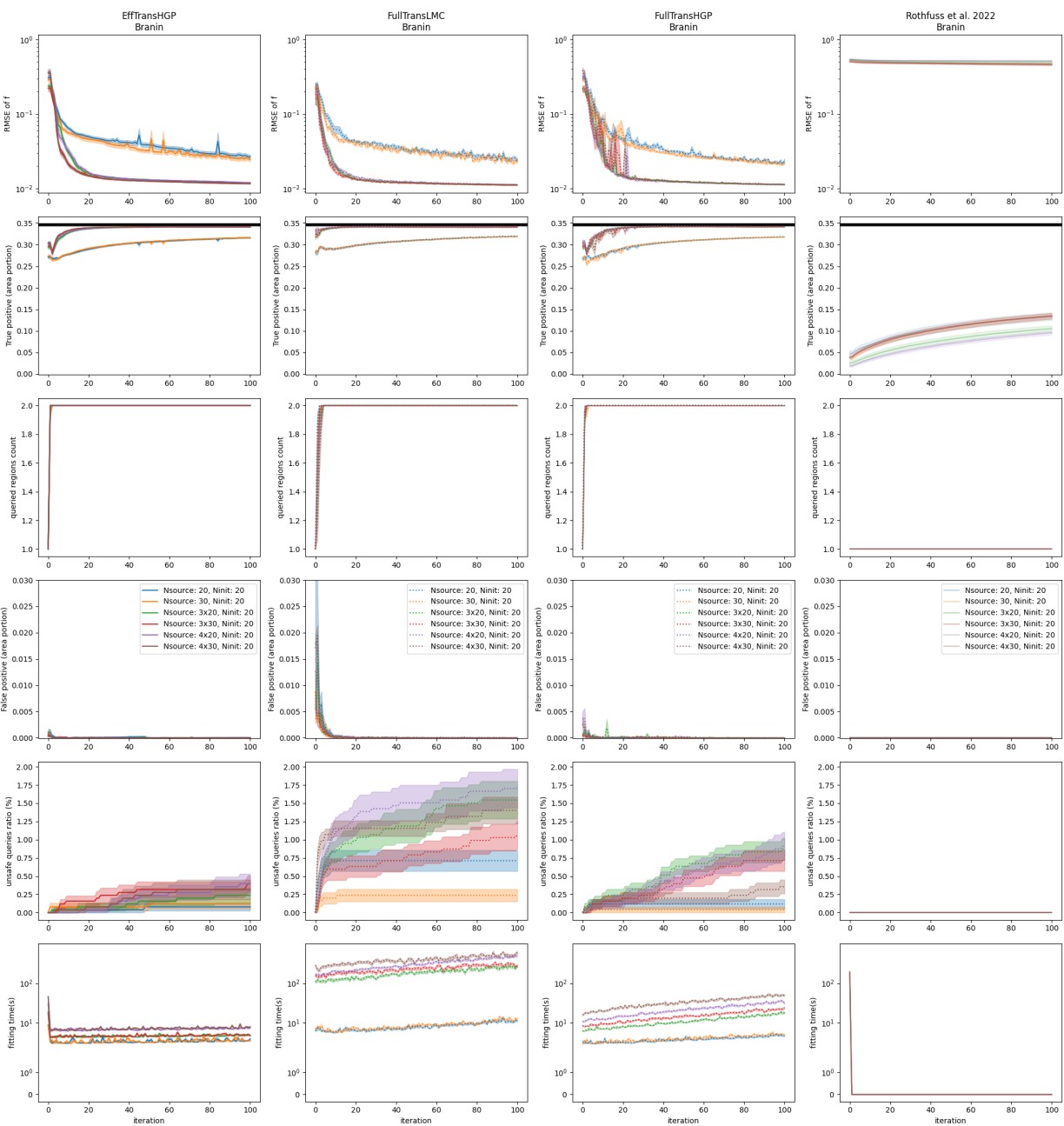

Figure 9: Safe AL experiments: Branin data with multiple source tasks. Each multi-task method is plotted in one column. We consider 1, 3 or 4 source tasks and sample 20 or 30 data points per task. The remaining setting is the same as described in Figure 8. RMSE plots are plotted in log scale.

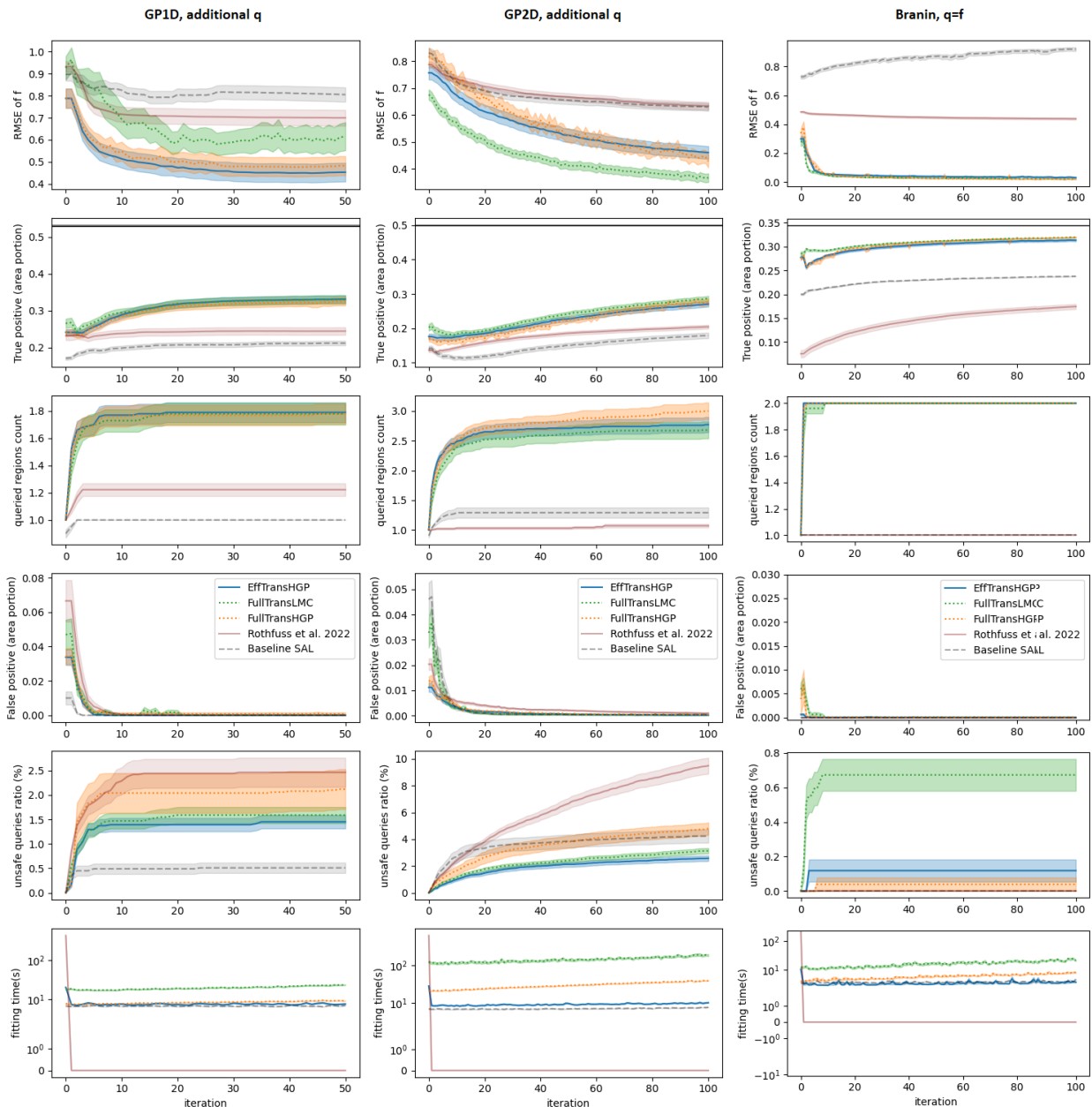

Figure 10: Safe AL experiments on three benchmark datasets: GP data with $\mathcal{X} = [-2, 2]^D$, $D = 1$ or 2, constrained to $q \geq 0$, and the benchmark Branin function with constraint $f \geq 0$. The results are mean and one standard error of 100 (GP data) or 25 (Branin data) experiments. $\mathcal{X}_{pool}$ is discretized from $\mathcal{X}$ with $N_{pool} = 5000$. We set $N_{source} = 100$ and $N$ is from 10 (0th iteration) to 60 (50th iteration) for GP1D, $N_{source} = 250$, $N$ is 20 to 120 for GP2D, and $N_{source} = 100$, $N$ is 20 to 120 for Branin. The first, second and fourth rows are presented in Figure 3 of the main paper. The TP/FP areas are computed as number of TP/FP points divided by $N_{pool}$ (i.e. TP/FP as portion of $\mathcal{X}_{pool}$). The third row shows the number of disjoint safe regions explored by the queries (main Table 2 is taken from the last iteration here). The fifth row, the unsafe queries ratio, are presented as percentage of number of iterations (e.g. at the 2nd-iteration out of a total of 50 iterations, one of the two queries is unsafe, then the ratio is 1 divided by 50). The last row demonstrates the model fitting time. At the first iteration (iter 0-th), this includes the time for fitting both the source components and the target components (EffTransHGP). With Rothfuss et al. 2022, source fitting is the meta learning phase.

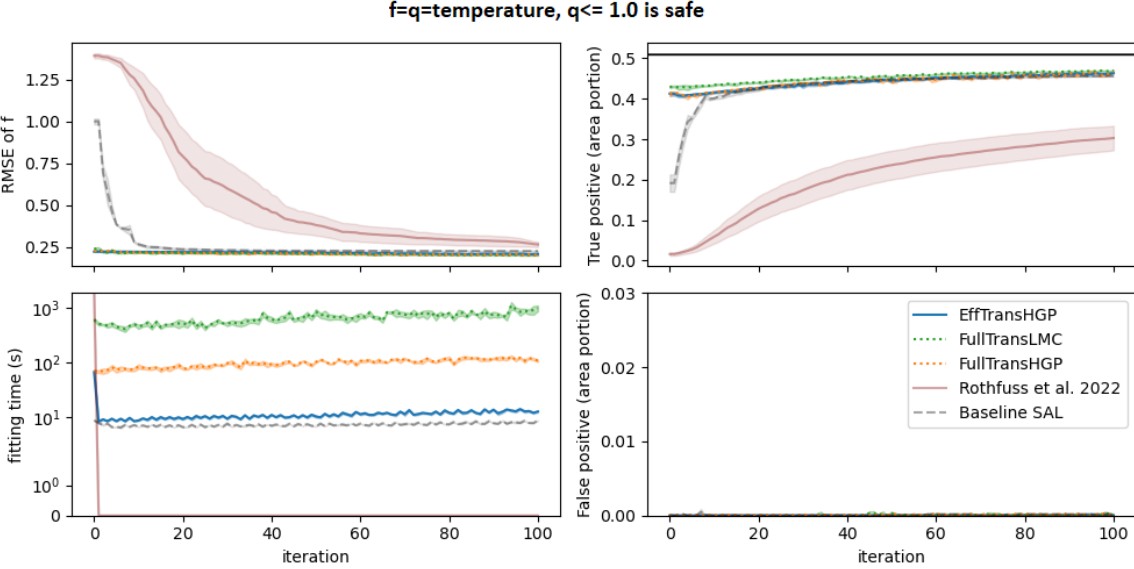

Figure 11: Safe AL experiments on engine emission modeling, AL on $f$ (temperature) constrained by $q = f \leq$ 1.0. Baseline is safe AL without source data. Transfer is LMC without modularization. Efficient_transfer is HGP with fixed and pre-computed source knowledge. $N_{source} = 500$, $N$ is from 20 to 120. The results are mean and one standard error of 5 repetitions. The fitting time is in seconds.

