# OpenReview forum: "Global Safe Sequential Learning via Efficient Knowledge Transfer"
_TMLR — Rejected by TMLR_

### Review · Reviewer_1vnd · 2024-03-03

**Summary Of Contributions:**

In this paper, the authors studied the problem of safe sequential learning where some source data are available. They propose a joint modeling approach incorporating source data to aid in predictive distribution in the target domain. Additionally, they propose a pre-computation technique to speed up the computation for posterior distributions. They also established some theoretical results on the limitation of local exploration of single-output Gaussian processes.

The submission includes empirical evaluations with both synthetic function data and engine modeling task data. Results show that the proposed algorithm can achieve lower MSE and identify more disconnected safe regions.

**Audience:**

Yes

**Claims And Evidence:**

Yes

**Requested Changes:**

1. In the section on “In-experiment speed-up via source pre-computation”, $\Omega_{g}$ is not defined (I think it refers to the covariance matrix being inverted).

2. In the caption to Figure 3, it is unclear to me what “$N$ is from 20 to 120” means. Since $N$ refers to the initial amount of target domain data, are you varying the amount of the initial target data here?

3. In Section 5.1, the authors stated “The datasets are generated such that the target task has at least two disjoint safe regions where each region has a common safe area shared with the source”. I found this condition for dataset construction too stringent. In practice, as source and target tasks can have different safety constraints altogether, this assumption seems to make the task too easy. Can the authors either justify this choice or include experiments that intentionally make the transfer from source to target domain more challenging?

**Strengths And Weaknesses:**

Strengths:
1. Safe exploration with Gaussian process models is a topic that is relevant to the TMLR audience. This work adds valuable theoretical and empirical understanding to this topic.

2. The proposed method is easy to understand and implement in practice. I found the innovation of using pre-computation for speed-up interesting.

Weaknesses:
1. I think the submission will benefit a lot by making the empirical evaluations more comprehensive. All the experiments are on low-dimensional datasets (<5 dimensions). Furthermore, including more datasets would make the algorithmic improvements more convincing as well.

2. It would be good to see an additional ablation study on the amount of the available source data and study its influence on the effectiveness of transfer learning.

3. There are some places where the descriptions are unclear. Please see the Requested Changes section below for details.

---

> ### Author Response · Authors · 2024-03-18
> **Ans to Reviewer 1vnd**
>
> ## Weak1.
> We include an additional datasets Hartmann3 (3 dim, section 5) and ablation studies on the amount of source data which is described below.
> The Hartmann3 dataset has been used previously in the community to test the efficiency of sequential data selection processes [1, 2]. Our results confirm that the performance increases (lower RMSE; higher safe area coverage) using our novel transfer learning approach.
>
> [1] Petru Tighineanu, Kathrin Skubch, Paul Baireuther, Attila Reiss, Felix Berkenkamp, Julia Vinogradska, AISTATS 2022, Transfer Learning with Gaussian Processes for Bayesian Optimization
>
> [2] Carl Hvarfner, Frank Hutter, Luigi Nardi, NeurIPS 2022, Joint Entropy Search For Maximally-Informed Bayesian Optimization
>
> ## Weak2.
> We agree that the amount of source data plays an important role in transfer learning.
> To study its effects in more detail, we ran additional experiments over the Branin datasets (appendix E).
> Our ablation study includes two new experiments: (i) we stay with one source tasks but vary the number of source data points, and (ii) we additionally vary the number of source tasks.
> The results of (i) show that the performance is increasing with more source data (lower RMSE, larger safe area coverage), and starting to saturate when the total number of source data points reaches around 100 points.
> Based on the results, we also modify the setting of Branin experiments in the main paper s.t. $N_{source}=100$ (originally $250$, mainly affect the fitting time, the remaining results are similar).
> With multiple source tasks, we also observe increasing performances which saturate with around 3 source tasks, 20 to 30 points per source task.
>
> ## Change1.
> Yes, thanks for pointing out.
> This was our mistake.
> We modify equation 4 accordingly.
>
> ## Change2.
> The initial number of target dataset is fixed.
> We were trying to indicate that the safe AL is operated from initially 20 points for 100 iterations, which result in 120 points in the end.
> To clarify this, we modify the notation, including algorithm 1&2, section 2, and experimental section: we introduce $N_{init}$ to explicitly indicate the number of initial target data, while N now indicates the number of target data which varies during AL.
>
> ## Change3.
> We add  clarification in section 5.
> In a transfer learning scenario, we always need information from the domain of interest.
> The Matern kernels (our base kernel) measure closeness of data points in the input space while our applied multi-output kernels rescale the base kernel measures for each task.
> This means the kernels used in the experiments correlate patterns of the same area in the input space, which is why we also generated datasets in the same way.
> Nevertheless, other base kernel choices may allow data points to be close in kernel space while distinct in input spaces.
> This becomes a problem of kernel selections, which is beyond the scope of this paper (see e.g.[3]).
> The multi-output kernels and our pipeline can remain the same.
>
> In addition, constrained source data is in fact a rather restrictive scenario.
> In a simulation-to-real example [4], the source data can be measured unconstrained, which provide information over the entire input space.
>
> [3] Matthias Bitzer, Mona Meister, and Christoph Zimmer, NeurIPS 2022, Structure Kernel Search via Bayesian Optimization and Symbolic Optimal Transport
>
> [4] Alonso Marco, Felix Berkenkamp, Philipp Hennig, Angela P. Schoellig, Andreas Krause, Stefan Schaal, Sebastian Trimpe, ICRA 2017, Virtual vs. real: Trading off simulations and physical experiments in reinforcement learning with bayesian optimization

---

### Review · Reviewer_Apea · 2024-03-04

**Summary Of Contributions:**

Sequential learning is essential in many applications. Two concrete (global) variants of it are active learning and Bayesian optimization. In practice, however, one has to pay attention to safety constraints on how to sample new data. Since it is often not trivial to identify feasible regions, the authors propose knowledge transfer from previous (source) learning tasks s.t. the performance on a target task can be improved. To this end, the authors describe how local exploration of feasible regions can be done with GPs (focussing on the active learning setting) and extend this to a modularized GP transfer learning for a given set of source tasks. In their experiments, they show that they can outperform previous approaches on 1D and 2D functions.

**Audience:**

Yes

**Claims And Evidence:**

No

**Requested Changes:**

1. Improve readability (see above)
2. Redo the experiments s.t. they have a practical relevance (see above).

**Strengths And Weaknesses:**

### Strengths:

1. The paper addresses a very important problem in practice. Since data acquisition is becoming more and more central to many AI applications, it is a very timely problem. In addition, the safety aspect is an understudied issue that is highly relevant in many applications.
2. The paper is very thorough on the theory side. It provides all formal background and assumptions and derives properties and bounds of the proposed approach.
3. The experiments are very well structured, along with meaningful research questions.

### Weaknesses:


1. The paper is not very well written. Since it provides a very deep theory and very good formalism, it is even more important to guide the reader through all the notation. After spending quite a bit of time, I was able to understand most of it – I admit that I was not able to check all the details. I fully believe that the authors know the notation by heart, but with all the subscripts and superscripts, I had to go back and forth several times to understand everything. It would make it much easier if the authors would give their notation more often a name s.t. one does not have to go back to the place where it was first introduced. Furthermore, a lot of important content is also hidden in the appendix. Even worse, some figures are in the appendix that are more marked as such in the text – making it painful to find them. Figure placement can be improved in general. In view of the recommended page limit of 12 pages, but only having 10 pages, I recommend improving readability substantially.
2. The benchmarks are only rather shallow.
 a. The authors only consider 1-D and 2-D functions. This is not of particular interest in practice, neither for active learning nor for Bayesian optimization.
 b. They use only a single source. Again, this makes the whole work very uninteresting for practical applications.
 c. Most benchmarks are artificial benchmarks (GP1D, GP2D and Branin).
 d. I don’t know whether the Engine datasets are publicly available s.t. one could reproduce the results. Furthermore, the Engine results are missing in Table 1.
 e. The authors wrote: “The GP model parameters are trained up-front and remain fixed during the experiments”. I would strongly argue that this is against common practice and needs clear justification as to why this is possible and reasonable.

Further questions to the authors:
 1. You only mention local safe learning. However, there are also prior works on global safe optimization. For example, Antonio Candelieri worked quite a bit on safe global optimization. Could you please comment on how this relates to your work?
 2. Could you please elaborate on X_pool. Why is this restriction needed?
 3. You said at the very end that a correlation between the source and the target is needed. This makes absolute sense, but what happens if this is not the case? Can your approach recover from it? If not, can you modify it so it can recover?

---

> ### Author Response · Authors · 2024-03-18
> **Ans to Reviewer Apea**
>
> ## Weak1.
> Thanks for pointing out.
> We try to reduce the subscripts and superscripts and we add table 1 to summarize our notation.
> We move the original fig 4 from appendix to fig 2 (in page 6, close to example 3.6 where it was referred).
> We make it clearer when we refer to appendix content.
>
> ## Weak2a.
> We include an additional dataset Hartmann3 (3 dim, section 5) and ablation studies on the amount of source data which is described below.
> The Hartmann3 dataset has been used previously in the community to test the efficiency of sequential data selection processes [1, 2]. Our results confirm that the performance increases (lower RMSE; higher safe area coverage) using our novel transfer learning approach.
> The safe learning literature with Gaussian processes commonly focuses on low dimensions (<= 5), e.g. [3, 4, 5].
> While we acknowledge this limitation of our and previous work, we nevertheless see practical relevance of previous and our work.
>
> [1] Petru Tighineanu, Kathrin Skubch, Paul Baireuther, Attila Reiss, Felix Berkenkamp, Julia Vinogradska, AISTATS 2022, Transfer Learning with Gaussian Processes for Bayesian Optimization
>
> [2] Carl Hvarfner, Frank Hutter, Luigi Nardi, NeurIPS 2022, Joint Entropy Search For Maximally-Informed Bayesian Optimization
>
> [3] Jens Schreiter, Duy Nguyen-Tuong, Mona Eberts, Bastian Bischoff, Heiner Markert & Marc Toussaint, ECML 2015, Safe Exploration for Active Learning with Gaussian Processes
>
> [4] Felix Berkenkamp, Angela P. Schoellig, Andreas Krause, ICRA 2016, Safe Controller Optimization for Quadrotors with Gaussian Processes
>
> [5] Christoph Zimmer, Mona Meister, Duy Nguyen-Tuong, NeurIPS 2018, Safe Active Learning for Time-Series Modeling with Gaussian Processes
>
> ## Weak2b.
> We agree that the amount of source data plays an important role in transfer learning.
> We ran additional experiments over the Branin datasets (appendix E).
> Due to space limit, we unfortunately have to refer to Weak2 of "Ans to Reviewer 1vnd" below for details.
>
> ## Weak2d.
> We will provide the engine dataset on github after publication.
> Engine experiments are not included in table 1 because the dataset does not have clearly separated disjoint safe regions, and because the connected component labeling algorithm is not always accurate due to noisy observation and grid interpolation (clarification at text below table 3).
>
> ## Weak2e.
> GP model parameters fixed up-front: this assumption was initially made in Sui et al. 2015, 2018, Rothfuss et al. 2022 etc and we took it over as it is.
> The papers provide theoretical results, that all the queries made by the algorithms are guaranteed to be safe with a certain probability. Such theorems hold true only when the same GP is used across all learning iterations.
> The idea behind this assumption is that GP parameters selected carefully by experts are safer than parameters fitted on the fly.
> In our experiments, the benchmark method Rothfuss et al. was taken over as it is, while our approaches (EffTransHGP, FullTransHGP, FullTransLMC) fit the parameters as the exploration proceeds.
>
> ## Weak, further1.
> We want to first point out that Bayesian optimization is sometimes called global optimization, as such methods aim to find global optima.
> However, under constraint(s), the search space is restricted, and we are only finding global optima “within” the identified safe region.
> Expanding the exploration to disjoint safe region requires safety forecasting into unseen region and this is, to the best of our knowledge, challenging.
> In [6], for example, the authors consider situation where the disjoint safe regions are separated by a small gap where the constraint function(s), with the noise, shortly goes beneath the safety threshold for a little bit (their figure 6).
> Their methods also have limit on identifiable regions (see their figure 13).
> We updated this in our related work.
>
> [6] Yaroslav D. Sergeyev, Antonio Candelieri, Dmitri E. Kvasov, Riccardo Perego, Soft Computing 2020, Safe global optimization of expensive noisy black-box functions in the delta-Lipschitz framework
>
> ## Weak, further2.
> In general, our algorithm also works without having X_pool.
> Due to space limit, we refer to the newly added remark 3.2 (page 5) for clarification.
>
> ## Weak, further3.
> We add clarification in section 5.
> In our particular pipeline, the Matern kernels measure closeness of data points in the input space while our applied multi-output kernels rescale the base kernel measures for each task.
> This means the kernels used in the experiments correlate pattern of the same area in the input space.
> However, it may happen that more sophisticated patterns are shared between tasks, which becomes a problem of kernel selection.
> The selection of kernel is itself a field of research and is beyond the scope of this paper. (see e.g.[7]).
>
> [7]  Matthias Bitzer, Mona Meister, and Christoph Zimmer, NeurIPS 2022, Structure Kernel Search via Bayesian Optimization and Symbolic Optimal Transport

---

> > ### Comment · Reviewer_Apea · 2024-03-20
> > **Space Limit?**
> >
> > Thanks for the revision and reply. Overall sounds promising, and I will look into the details soon. In the meantime, I wondered why you argue with a space limit. There is no space limit for TMLR; if you have more than 12 pages, the review needs more time, but the initial review is already done. So I see no reason not to go for a fully polished paper without concerns about a space limit.

---

> ### Author Response · Authors · 2024-03-20
> **space limit**
>
> Thanks for the feedback.
>
> The open review seems to restrict the max character count in each comment.
> We cut few sentences in the answers which also appears somewhere else (but the paper is in its full version).
>
> Sorry, I actually realized now that we could have split the reply into two comments.
>
> For now, we keep it unchanged.
>
> If anything is still unclear, we are happy to answer again!

---

### Review · Reviewer_87Xf · 2024-03-05

**Summary Of Contributions:**

This paper presents a multi-output GP approach to address the safe exploration challenge in active learning. The key idea is to enable knowledge transfer from a related source so that the safe region can be estimated better (i.e., requiring fewer prior data from the target task).

In this setup, the target task is characterized by one GP-distributed main function and a set of GP-distributed constraint functions. At each iteration, an input candidate (whose constraint outputs are above a pre-specified threshold) with maximum predictive entropy will be selected. All GPs will be retrained for the next iteration and so on.

The main contributions of this paper include: (1) the derivation of the safety probability in the single-output GP (no transfer); and (2) the use of multi-output GP to enable transfer from an existing source to the target problem. The source is represented in a similar manner to the target where there is a GP-distributed (source) main function and several other GP-distributed constraint functions.

**Audience:**

Yes

**Claims And Evidence:**

Yes

**Requested Changes:**

Please address the concerns that I raised in the Weakness section above.

**Strengths And Weaknesses:**

Strengths:

The paper is well-organized and well-written.
The idea of source transfer to boost the accuracy of the safety probability estimate is sufficiently novel.
The technical flow is clear and sound.

Weaknesses:

1. The result in Theorem 3.3 is giving out a vibe that it is trivial and can also be come vacuous (due to strong independence assumption). To elaborate:

As the predictive distribution is a Gaussian, the chance that the prediction being larger than a threshold is always represented in forms of a Gaussian cumulative function and the rest of the algebraic inside the cumulative function are direct consequences of the assumption.

Furthermore, assuming the constraint functions are independent will lead to a vacuous bound. Even if the safety probability per constraint is sufficiently close to 1.0, their product will approach zero when there are sufficiently many constraints -- see the text after Eq. (2).

This suggests that the technical setting here might be oversimplified. In practice, constraint functions are likely correlated. I think the authors might want to elaborate more on this point if I have missed something important here.

2. The paper also gives the vibe that the above theoretical guarantee does not extend to the source-transfer setting via multi-output GP. Can the authors provide clarification on this? The multi-output GP is a GP with co-regionalized kernel so I tend to think any results for single-output GP would extend straightforwardly to multi-output setting.

3. On the same thought that the multi-output GP is a GP with co-regionalized kernel, I wonder why the authors prefer inventing new approximation to adopting existing literature on sparse GP (to improve scalability). Please elaborate.

4. Several technical details are not quite clear. For example, how do we maximize the acquisition function? how do we guarantee that the maxima of the acquisition function will remain within the safety set? How do we define the co-regionalization kernel in cases where both source and target have multiple constraint functions. The paper seems to detail only the formulation when there are only two constraint functions (one for source, one for target). It would be good to flesh this out clearer.

5. Experiments are mostly conducted on simulated or toy datasets in low dimensions. The authors should probably consider running their proposed solution on more benchmark datasets with larger dimension and quantity.

6. I also do not see concrete theoretical analysis showing the sample save-up in target domain when there is a source transfer. Can the authors discuss this?

7. The notation is probably heavy to people who are not already familiar with GP. Perhaps the authors should consider lightening up the notation.

---

> ### Author Response · Authors · 2024-03-18
> **Ans to Reviewer 87Xf**
>
> ## Weak1.
> We thank the reviewer for raising this point.
> We agree that in the limit of sufficiently many constraints the set $\mathcal{S}_N$ could become empty.
> However, in practice it is relatively uncommon to have this many (asymptotical scenario) constraints.
> If users are worried about the safe set getting too small, modeling different constraints with multioutput GP does not contradict any technical assumption of our pipeline.
> In this case, we could concatenate a number of target constraints and the corresponding source data, and then we model the concatenation with one multi-output GP.
> The source pre-computation (our algorithm 2) may still be applied by fixing the block of all the sources.
> We added remark 4.2 in section 4 for this.
>
> ## Weak2.
> We want to clarify that our theoretical analysis provides an explicit bound to the explorable region given the observations.
> The key insight here is that we can show when single task modeling is absolutely not effective.
> While the same result holds for multitask GP, all of the source data needs to be considered together with the target initial data.
> With large amount of source data, such explorable bound is significantly enlarged comparing to modeling solely with the target task.
>
> ## Weak3.
> We do not approximate the GP distributions (e.g. as a sparse GP does).
> Such safe learning methods are usually performed on low data regime, where we can afford computing the full GP.
> We only want to avoid unnecessary computation.
> To give a concrete example of the benefit: if we have 500 source data and we query with 20 initial target data, then algorithm 1 for 100 iterations means inverting a gram matrix of size 520x520, then inverting a matrix of size 521x521 and so on.
> This means an overall around 14 hours (our main engine example, FullTransferLMC) or 2.5 hours (FullTransferHGP) acquisition time, in contrast to EffTransHGP where 100 iterations take less than 20 minutes.
> In many engineering examples, including our engine task, keeping the machine active for hours or days v.s. for less than 20 minutes matters significantly in cost and effort.
>
> ## Weak4.
> We solve this by considering only a discrete pool with finite number of elements.
> This means, we compute the GP predictive distributions (equation 1 and 4) over the entire pool to determine the safe set (equation 2), then filter out unsafe candidates, compute the acquisition function over the safe points, and finally pick the point with the largest acquisition score.
> We add remark 3.2 into the paper for clarification.
> Regarding the safety functions, each constraint is considered independently (equation 2) and the safe set is the set with all safety constraints satisfied.
> The correlation, in our case, considers only the same constraint of the source and the target.
> For example, one may consider the engine temperature as a constraint, correlating a well-measured source engine and a newly developed target prototype. If we consider e.g. engine torque as another constraint, then $q_1$ is temperature (multitask, source temperature and target temperature), and $q_2$ is the torque (multitask, source torque and target torque). $q_1$ and $q_2$ are modeled independently.
>
> ## Weak5.
> We include an additional dataset Hartmann3 (3 dim, section 5) and ablation studies on the amount of source data (appendix E).
> The Hartmann3 dataset has been used previously in the community to test the efficiency of sequential data selection processes [1, 2].
> Our results confirm that the performance increases (lower RMSE; higher safe area coverage) using our novel transfer learning approach.
> Furthermore, we ran the following ablation experiments on Branin datasets (appendix E): (i) we stay with one source tasks but vary the number of source data points, and (ii) we additionally vary the number of source tasks.
> The results show that the performance is increasing with more source data (lower RMSE, larger safe area coverage), and starting to saturate when the total number of source data points reaches around 100 points (or with around 3 source tasks, 20 to 30 points per source task).
> Based on the results, we also modify the setting of Branin experiments in the main paper s.t. $N_{source}=100$ (originally $250$, mainly affect the fitting time, the remaining results are similar).
>
> [1] Petru Tighineanu, Kathrin Skubch, Paul Baireuther, Attila Reiss, Felix Berkenkamp, Julia Vinogradska, AISTATS 2022, Transfer Learning with Gaussian Processes for Bayesian Optimization
>
> [2] Carl Hvarfner, Frank Hutter, Luigi Nardi, NeurIPS 2022, Joint Entropy Search For Maximally-Informed Bayesian OptimizationWeak6. This is only demonstrated empirically (RMSE decreases, safe set coverage increases).
>
> ## Weak6.
> This is only demonstrated empirically (RMSE decreases, safe set coverage increases).
>
> ## Weak7.
> Thanks for pointing out. We try to reduce the subscripts and superscripts and we add a table (table 1) summarizing our notation.

---

### Author Response · Authors · 2024-03-18
**Summary rebuttal**

We thank the reviewers for their encouraging feedback that "Safe exploration with Gaussian process models is a topic that is relevant to the TMLR audience” (Rev 1vnd), that “the safety aspect is an understudied issue that is highly relevant in many applications” (Rev Apea) and our “proposed method is easy to understand and implement in practice” (Rev 1vnd).
We also thank the reviewers for pointing out improvement. In the two weeks rebuttal, we

a)	added an ablation study on the number of source data (see appendix Fig 8),

b)	added results on the effect of the number of source tasks (different from a where we modified the number of points of one source task, appendix Fig 9),

c)	added an additional data set, Hartmann 3, which is of a higher dimension,

d)	clarified notation and added Table 1 with notation overview.

We will give a detailed point-by-point response below for each reviewer individually.
If you have more questions, feel free to give us further feedback. We are happy to get back to the concerns.

---

### Decision · Action_Editor_4iQi · 2024-04-09

**Recommendation:** Reject

**Comment:**

This paper studies a transfer learning problem where the safety constraints are also transferred. The transfer is done using Gaussian processes. The original paper had several shortcomings, such as hard to read notation and experiments on mostly toy datasets. The authors updated the paper significantly. The notation simplified and is also summarized in Table 1, which improves readability. However, some issues, such as mostly toy datasets, remained. Therefore, the paper cannot be accepted at this time.

These are additional comments of the reviewers from the recommendation to accept / reject. I hope that you will find them useful:

**Reviewer Apea**

The clarity of the paper could still be improved, and the benchmarks are also weak, in particular the low-dimensional, artificial functions worry me. I'm also still concerned regarding the fixed GP parameters. I see reasons why prior work also did it that way, but I nevertheless believe that this prevents any reasonable insights into practical applications of the proposed approach.

**Reviewer 87Xf**

First, regarding weakness #1, I agree with the authors that when the constraints are not independent (as assumed), a potential fix is to model them using a multi-output GP which explicitly characterizes their correlation. But, the rebuttal does not address how that would affect their theoretical results which might be built on that assumption. The added text in Remark 4.2 does not seem to elaborate more on this. Hence, while I believe this is a potential fix but my impression is that the authors have not quite fleshed it out.

Second, regarding weakness #3, I have the impression that the authors had misunderstood my question or maybe I simply do not understand the main point of the response. I was asking that in the spirit of speeding up the GP computation, why wouldn't the authors simply adopt sparse GPs to achieve this instead of leveraging prior pre-computation, which is not quite as effective.

Certainly, if the combined data from both source and target are small, this should not be an issue. But, I tend to think that that should not be the case here since the authors' response to weakness #2 indicated that the combined source data is expected to be large. In this case, this might be a problem because the complexity of the pre-computed approach devised by the authors is quadratic in the amount of source data.

Third, regarding weakness #4, the authors revealed that the maximization of the acquisition function is based on a discretization. But, the detail regarding the discretization is not quite clear, e.g., will it impact the derived guarantee? how will it impact the computational cost, especially when the discretized mesh can grow exponentially in the no. of input dimensions?

Last, regarding weakness #5, even with the addition of the Hartman dataset, it is still the case that the empirical studies have been conducted mostly on very low-dimensional (toy) datasets. Other than the obvious need to demonstrate the practical significance of the algorithm on more sophisticated, high-dimensional datasets. there is also a concern that arises from the fact that the acquisition function is optimized based on an exhaustively enumeration over a discretized mesh. Will it still be scalable on high-dimensional domains?

**Audience:**

Yes. This paper would appeal to the Bayesian optimization community. It brings new ideas for transferring safety constraints.

**Claims And Evidence:**

No. One issue that was brought up by all reviewers is that most experiments are on toy datasets with up to 3 dimensions: GP1D, GP2D, Branin, and Hartmann3. Hartmann3 was added during the rebuttal and is also a toy dataset. This issue needs to be addressed before the paper is submitted to TMLR again.

**Resubmission Of Major Revision:**

The authors may consider submitting a major revision at a later time.